# ATAC: Abstractive Token-Level Question-Agnostic Prompt Cmprsr

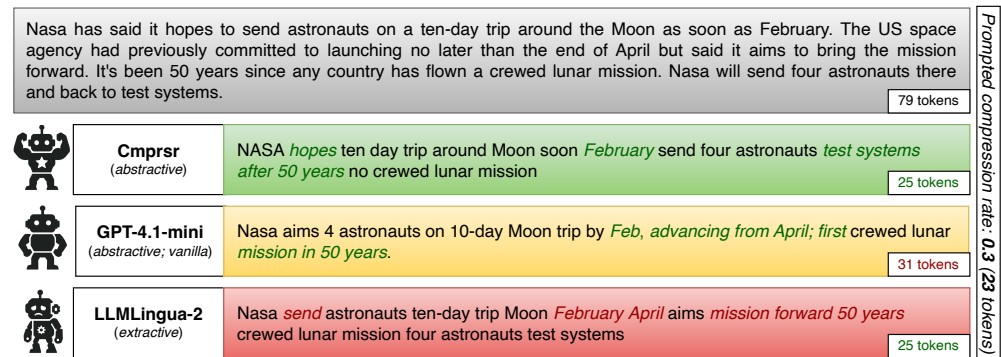

Figure 1: Extractive compression selects a subset of the input sequence tokens, while abstractive compression allows for clever paraphrases. While one can use vanilla LLMs as abstractive compressors, their performance can be further improved with RL-based post-training, yielding **Cmprsr**. Note that extractive compression may introduce ambiguities, *e.g.* "February April", "mission forward 50 years", while vanilla abstractive compression does not adhere to the desired compression rate. The original text snippet is from BBC (2025).

## ABSTRACT

Motivated by the high costs of using black-box Large Language Models (LLMs), we introduce a novel prompt compression paradigm, under which we use smaller LLMs to compress inputs for the larger ones. We present the first comprehensive LLM-as-a-compressor benchmark spanning 25 open- and closed-source models, which reveals significant disparity in models' compression ability in terms of (i) preserving semantically important information (ii) following the user-provided compression rate (CR). We further improve the performance of *gpt-4.1-mini*, the best overall vanilla compressor, with *Textgrad*-based compression meta-prompt optimization. We also identify the most promising open-source vanilla LLM—*Qwen3-4B*—and post-train it with a combination of *supervised fine-tuning (SFT)* and *Group Relative Policy Optimization (GRPO)*, pursuing the dual objective of CR adherence and maximizing the downstream task performance. We call the resulting model **Cmprsr** and demonstrate its superiority over both extractive and vanilla abstractive compression across the entire range of compression ratios on lengthy inputs from *MeetingBank* and *LongBench* as well as short prompts from *GSM8k*. The latter highlights **Cmprsr**'s stable performance for varying input types. Moreover, **Cmprsr** closely follows the requested compression ratio, offering fine control over the cost-quality trade-off.

## 1 ABSTRACTIVE COMPRESSION WITH LLMs

The discovery of scaling laws Kaplan et al. (2020) set the trend for training increasingly large Language Models (LMs). Despite rapid advancements in both hardware and software supporting LLMs'

inference, the costs of their usage continue to surge. This trend reflects not only growing adoption Liang et al. (2025), but also the Jevons paradox Luccioni et al. (2025): efficiency gains that spur even greater consumption.

According to recent estimates Tully et al. (2025), $87\%$ of enterprise spendings on LLMs are attributed to the black-box LLMs, accessed via API. This means that in practice the only way the majority of LLM consumers can optimize their spendings is through minimizing the length of the queries they pass to the models; this can be achieved with the *token-level prompt compression* Li et al. (2025), *i.e.* exploiting the redundancy of the human language Shannon (1951) and mapping the original input sequence to a shorter one, while preserving original semantics.

We focus on the *question-agnostic compression* Jiang et al. (2023), which aims to process the context provided to the model so that the compression can be re-used for any context-related question or task: possible use-cases include (i) combining compression with Retrieval-Augmented Generation Gao et al. (2023) via retrieving pre-compressed entries, (ii) single-call compressions of lengthy texts such as meeting transcripts Hu et al. (2023) for their subsequent comprehensive analysis involving multiple LLM calls, (iii) optimizing LLM-powered learning platforms via compressing learning materials (iv) addressing tokens-per-minute (TPM) API bandwidth limitations.

The most popular approaches tailored for this set-up are **extractive** Jiang et al. (2023); Pan et al. (2024), meaning that compression is posed as a binary classification problem of preserving/removing each of the input sequence parts (usually tokens). We hypothesize that **abstractive** methods, operating under a much larger space of valid compressions, can provide better outputs through paraphrase of the input sequence. To this end, we define 2 metrics, characterizing the quality of a *Compressor* LLM: (i) *Target* LLM performance on a downstream task given its inputs are preprocessed by the *Compressor* (ii) Adherence of the *Compressor* to the user-defined *Compression Rate (CR)*, reflecting user's tolerance to the quality deterioration versus the incurred costs. We ask the following research questions (RQs):

- **RQ1**. What are the compression capabilities of the off-the-shelf LLMs?
- **RQ2**. Can we further improve their performance with prompt optimization techniques such as Yuksekgonul et al. (2025)?
- **RQ3** How does performance of an SFT/RL post-trained "small LLM" *Compressor* compares to the SOTA **extractive** approaches across different datasets?

We share results of the extensive benchmarking addressing **RQ1**, and show that the answer to **RQ2** is positive in case the prompt is used with the same *Compressor* model it was optimized for. Most importantly, we present abstractive **Cmprsr**, outperforming SOTA extractive compression across different compression rates, which answers **RQ3**.

## 2 RELATED WORK

Compression can be performed in either *question-aware* Shandilya et al. (2024); Kim et al. (2025); Choi et al. (2024); Yoon et al. (2024) or *question-agnostic* way. In the *question-aware* set-up, compression is conditioned on the given question and aims to filter out all irrelevant information. While allowing high compression rates, this also prevents compression re-usage for the new queries. Motivated by the use-cases detailed in Sec. 1, we focus on the *question-agnostic* set-up.

Furthermore, prompt compression techniques can be divided into 2 categories Li et al. (2025):

**Embedding-level** (*a.k.a* soft prompt methods Mu et al. (2023); Chevalier et al. (2023)), which require access to the internals of the *Target* model, and are therefore not viable for the black-box models.

**Token-level** (*a.k.a* hard prompt methods): (i) *Extractive*: compression space is the set of order-preserving subsequences of the input; the filtering can be implemented on the token Jiang et al. (2023); Pan et al. (2024) or the sentence Liskavets et al. (2025) level. Notably, Hu et al. (2025); Shandilya et al. (2024) use RL techniques to optimize the extraction policy. (ii) *Abstractive*: the compression space is all sequences over the vocabulary, which allows semantics-preserving paraphrase/reordering via tokens not present in the input.

Below, we detail prior contributions falling into the same broad category as ours, *i.e.* **question-agnostic token-level abstractive compression**. Pu et al. (2024) rely on vanilla *LLaMA-2-7B* to perform compression guided by demonstrations, optimized for the particular dataset. Despite improvement over straightforward prompting, this method is not readily generalizable across tasks, as each new dataset requires generating/selecting a new set of demonstrations. In a recent work, Zhang et al. (2025) develop another approach to abstractive compression, using either *GPT-4o-mini* or *Qwen-2.5*: they rely on dynamic chunking and derivation of chunk-specific compression ratios, which allows to preserve important context.

Neither of the two methods above compares a meaningful number of LLMs in terms of their aptitude for compression, which, as we show, greatly varies. Furthermore, according to our experiments, even the models dominating "vanilla compression leaderboard" can be significantly improved with post-training in terms of both compression quality and adherence to the user-specified compression rate. The work of Chuang et al. (2024) is the one we find closest to addressing this limitation: while it still experiments with a single backbone (*Vicuna-7B*), the model is actually tuned for better compressions. The main limitation of the presented approach is that it falls short of adopting RL: *Vicuna-7B* serves as both *Compressor* and the *Target* model, and the signal comes from the semantic preservation loss between the original and the compressed input activations. Moreover, the desired absolute length of the compression is ingrained in the loss during the training stage and is thus non-adjustable during the inference time.

Among the works targeting specific downstream tasks, Larionov & Eger (2025) investigate compression for the machine translation quality assessment. While the whole training pipeline is tailored for this task and the resulting model cannot be used for prompt compression "in the wild", the authors notably introduce RL —in the form of preference optimization (ORPO)—for training an abstractive compressor.

## 3 COMPRESSIONBENCH

We focus on the following 2 metrics reflecting the practical usability of a *Compressor* model: (i) adherence to the desired compression ratio: $CR = n_{tkns}^{cmpr}/n_{tkns}^{original}$; $\Delta_{CR} = CR_{real} - CR_{desired}$; (ii) performance on the downstream tasks given compressed inputs. In case of MeetingBank (MB), 2 downstream tasks are summarization and question-answering (see Fig. 2). The template of the *Compressor* prompt, including the length conditioning—we render compression rate into the desired number of tokens and add it to the system prompt—is given in the Appendix. Both here and when training **Cmprsr**, we cut transcripts into chunks before passing them to the *Compressor* model, following the methodology of Pan et al. (2024) to avoid truncated final sentences in the produced chunks. We then combine compressed chunks back into the compressed transcripts.

We present the most important MB results in Table 1, and provide full MB results (Table 4) along with the GSM8k results (Table 5) and the full names of the models (Table 3) in the Appendix. We dissect the benchmarking results below as a set of enumerated **findings**: F1, F2, F3, and F4.

**F1 Vanilla LLMs poorly adhere to the prompted compression rate.** While they are susceptible to the prompted rate, the length of the generations skews towards some fixed CR, leading to "under-compression" for high compression rates (*0.1*) and "over-compression" for low compression rates (*0.5*). *LLMLingua-2* does not suffer from this limitation, as the classification threshold can be dynamically adjusted for the extractive methods.

**F2 Abstractive LLMs excel at high compression rates**. Unlike extractive methods, they can rephrase and condense information beyond the original tokens, preserving key semantics. This highlights the crucial role of abstraction for aggressive compression.

**F3 Comparison of LLMs.** Although closed-source models generally outperform large open-source models, which in turn surpass smaller open-source ones, performance is not monotonic within each class. Model size or release date alone does not predict the outcome: for example, *gpt-4.1-mini* outperforms both *gpt-4.1* and *gpt-5-mini* on compression ratio (CR) adherence and compression quality.

**F4 LLMs vastly outperform *LLMLingua-2* on the shorter prompts from GSM8k**, although CR adherence is worse than for long MB prompt (Table 5).

Table 1: Compression performance of various vanilla models on the MeetingBank transcripts: truncated version of Table 4. Within each group, we sort the models based on the average QA performance accross compression rates.

| | $\Delta_{CR}$ | | | BERT-F1 | | | QA | | |
|---|---|---|---|---|---|---|---|---|---|
| Requested CR | 0.1 | 0.3 | 0.5 | 0.1 | 0.3 | 0.5 | 0.1 | 0.3 | 0.5 |
| **Closed-source models** | | | | | | | | | |
| gpt-5-nano | 0.19 | 0.17 | 0.04 | 0.87 | 0.88 | 0.88 | 0.25 | 0.30 | 0.31 |
| gpt-4.1-mini | 0.07 | -0.00 | -0.17 | 0.89 | 0.90 | 0.90 | 0.20 | 0.29 | 0.30 |
| gpt-5-mini | 0.13 | 0.11 | 0.10 | 0.86 | 0.87 | 0.87 | 0.19 | 0.25 | 0.27 |
| gpt-5 | 0.10 | 0.08 | 0.06 | 0.87 | 0.88 | 0.88 | 0.18 | 0.25 | 0.25 |
| gemini-2.0-flash-lite | 0.09 | 0.00 | -0.17 | 0.88 | 0.89 | 0.89 | 0.17 | 0.24 | 0.25 |
| gpt-4.1-nano | 0.04 | -0.02 | -0.19 | 0.87 | 0.89 | 0.89 | 0.15 | 0.26 | 0.25 |
| gpt-4.1 | 0.06 | -0.02 | -0.19 | 0.88 | 0.89 | 0.89 | 0.17 | 0.23 | 0.25 |
| gemini-2.5-flash | 0.03 | -0.03 | -0.16 | 0.88 | 0.89 | 0.89 | 0.15 | 0.22 | 0.24 |
| o4-mini | 0.10 | 0.05 | -0.17 | 0.88 | 0.88 | 0.88 | 0.14 | 0.22 | 0.21 |
| **Large Open-source models ($> 10B$)** | | | | | | | | | |
| gemma-3-12b-it | 0.18 | 0.07 | -0.05 | 0.89 | 0.89 | 0.89 | 0.22 | 0.25 | 0.26 |
| Mistral-Small-3.1-24B | 0.14 | -0.04 | -0.23 | 0.88 | 0.89 | 0.89 | 0.22 | 0.24 | 0.23 |
| DeepSeek-V3 | 0.11 | -0.05 | -0.24 | 0.89 | 0.89 | 0.89 | 0.21 | 0.24 | 0.25 |
| **Small Open-source models ($< 10B$)** | | | | | | | | | |
| Llama-3.2-3B | 0.05 | -0.10 | -0.30 | 0.87 | 0.87 | 0.87 | 0.17 | 0.21 | 0.22 |
| Qwen3-4B | 0.05 | -0.08 | -0.26 | 0.86 | 0.88 | 0.88 | 0.16 | 0.21 | 0.22 |
| Qwen2.5-7B | 0.08 | -0.11 | -0.31 | 0.87 | 0.88 | 0.88 | 0.16 | 0.19 | 0.18 |
| gemma-3-4b | 0.00 | -0.15 | -0.35 | 0.87 | 0.87 | 0.88 | 0.13 | 0.17 | 0.19 |
| Llama-3.1-8B | -0.02 | -0.19 | -0.39 | 0.85 | 0.86 | 0.86 | 0.14 | 0.17 | 0.15 |
| Qwen2.5-3B | 0.09 | -0.08 | -0.24 | 0.85 | 0.86 | 0.86 | 0.10 | 0.12 | 0.12 |
| **Extractive** | | | | | | | | | |
| llmlingua-2 | -0.01 | -0.03 | -0.03 | 0.86 | 0.89 | 0.9 | 0.16 | 0.34 | 0.42 |

The results from this section inform our choice of the models for the **Cmprsr** experiments: *Qwen3-4B* among the small open-source models (it performs on par with *LLams-3.2-3B* with slighly better CR adherence), and *gpt-4.1-mini* among the closed-source ones.

## 3.1 BOOSTING VANILLA PERFORMANCE WITH TEXTGRAD

**Motivation.** The system prompt is crucial for aligning LLM output with a user's expectations. As a result, it has a major impact on the performance of our LLM-based compressor. With the increase in LLM popularity, prompting has grown into a mature field with a variety of techniques, including step-by-step reasoning instructions and few-shot examples. However, choosing *which* technique to use and *how* to phrase it remains nontrivial and highly task-dependent. To solve both of these problems, we use TEXTGRAD Yuksekgonul et al. (2024) for principled prompt optimization. TEXTGRAD treats textual components of compound LLM systems as optimization variables and improves them via natural-language "gradients."

**Method in brief.** We model the MeetingBank QA benchmark as a computation graph with nodes corresponding to stages in the LLM pipeline. Each node is assigned a role description, helping the optimizer-LLM understand the high-level interdependencies among nodes. Starting from the ground-truth–based evaluation at the output, TEXTGRAD (i) identifies failure modes, (ii) generates suggestions on how to fix them, (iii) *backpropagates* this feedback through the graph to the upstream nodes, and (iv) uses the accumulated feedback to update the compressor system prompt. For evaluation, we use a 10-point LLM Judge score to assess QA performance, where the LLM is asked to evaluate each answer and assign a score. More Details on TEXTGRAD are provided in Appendix B.

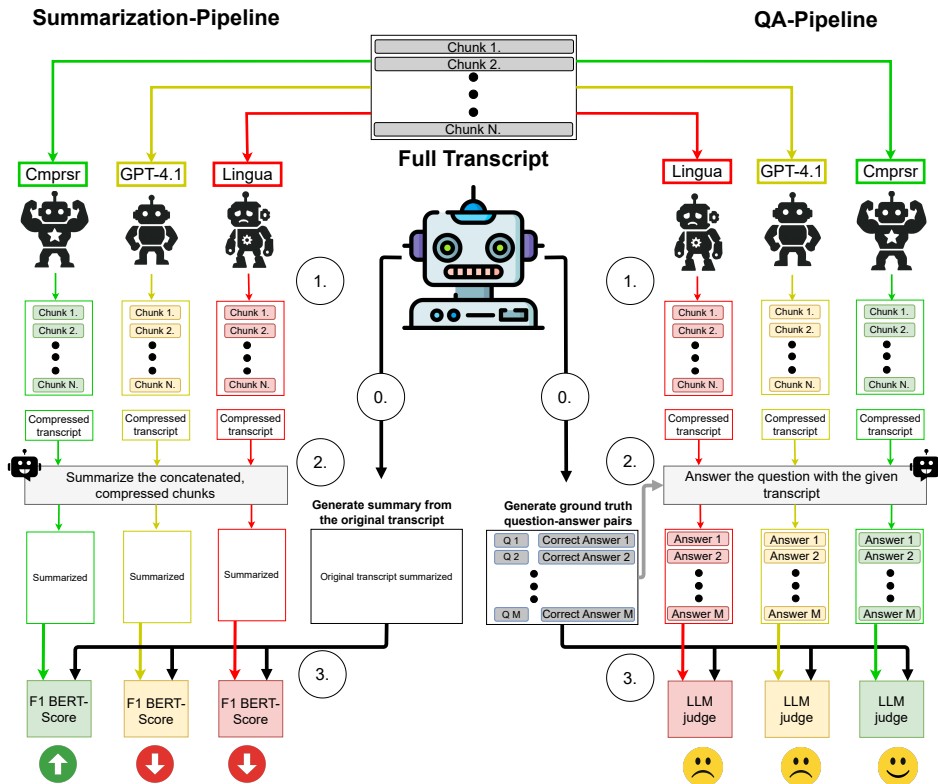

Figure 2: MeetingBank evaluation pipeline. We assess transcripts' compressions on 2 downstream tasks. (i) **Summarization**, i.e we compute BertScore between compressed and original. (ii) **QA**, where we build a dataset of questions and answers from MeetingBank transcripts, and measure the *Target* model's accuracy using the compressed context.

**Results.** As TEXTGRAD iteratively updates the prompt, the model outputs become increasingly better aligned with the downstream QA task. Figure 3 illustrates the tradeoff between the QA quality and adherence to the CR across iterations, and shows excerpts from the initial prompt and the best quality prompt (iteration 8). The optimized prompt yields a +0.51 gain in average LLM Judge score, with only a -0.02 drop in CR adherence. Interestingly, the learned prompt addresses the uncovered failure modes by adding a positive and a negative example as well as by stressing the importance of named entities and numeric values. As a result, the updated prompt steers the compressor to retain information most salient for answering questions while remaining close to the desired token budget.

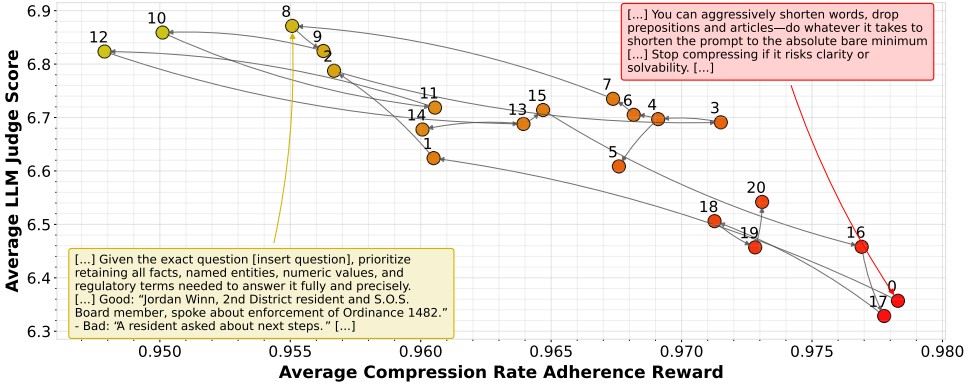

Figure 3: TextGrad system prompt optimization for Qwen3-4B on MeetingBank QA dataset.

In Figure 3, each point is an iteration of TEXTGRAD; the $x$-axis shows the average compression-rate adherence reward (higher is better), and the $y$-axis displays the average LLM-Judge QA score (0–10, higher is better), both computed on a 100-transcript hold-out set. Panels include parts of the initial system prompt (iter. 0) and the prompt with the best quality (iter. 8).

# 4 CMPRSR

## 4.1 SUPERVISED FINE-TUNING (SFT)

**Data generation.** We use the strongest vanilla *Compressor—gpt-4.1-mini—*to generate *MB* compressions for the subsequent distillation, while randomly sampling prompted compression rate from the $[0.1, 0.7]$ range. We notice that the distribution of lengths of the generated data points does not follow the uniform distribution of the prompted rates, skewing to $\approx 0.3$. In order to mitigate that, we experiment with 2 re-balancing strategies: (i) downsampling, *i.e.,* dropping entries with over-represented lengths, (ii) *upsamplig*, *i.e.,* boosting under-represented ones. We label each generated compression with the number of tokens it contains.

**SFT.** The aim of this stage is twofold: (i) improving *Qwen3-4B* performance through distilling the compression strategies of *gpt-4.1-mini* (ii) improving adherence to the prompted compression rate. To this end, we append the true number of tokens in the compression to the prompt, i.e., the structure of the sequence the fine-tuning is performed on is `Original + Length Conditioned Prompt (len(Compression)) + Compression`, and the loss is computed on the `Compression` part; this "length conditioning" is inspired by the Hindsight Instruction Relabelling (*HIR*) works Zhang et al. (2023); Shypula et al. (2024). In the preliminary experiments, we also tested additional "quality conditioning", *i.e.* passing the normalized quality of the compression relative to other compressions in the same compression rate range. This research direction, however, had limited success. We varied ($lr$) in the $[10^{-4}, 10^{-8}]$ range, and picked SFT checkpoint with the best validation accuracy, which is the one, trained on the upsampled dataset with $lr = 10^{-5}$.

## 4.2 GRPO

To ensure that the model not only achieves the desired compression ratio but also retains task-relevant information, we extend the SFT stage with GRPO training. For each input chunk $x$ and its compressed form $x_C$, we define two complementary reward functions: a *length reward* to control compression and a *quality reward* to maintain semantic fidelity.

**Length reward.** The length reward (1) encourages the model to adhere to the target compression ratio $r_T$. For a given input, the achieved compression ratio is computed as $r_C = \frac{|x_C|}{|x|}$. This formulation penalizes over-compression (when $r_C > r_T$), while assigning a reward close to 1 if the compression stays within or below the target threshold.

$$R_{\text{len}} = 1 - \max\big(0, r_C - r_T\big), \tag{1}$$

**Quality reward.** The **quality reward** $R_{\text{qual}}$ (2) measures how well the compressed chunk $x_C$ preserves information relevant to the downstream task. We initially experimented with a *QA-based reward*, i.e., the reward is computed by evaluating the compressed text on a QA task against the ground-truth answer from the uncompressed text. However, we found the model exploited this reward by outputting many possible answers instead of compressing effectively. We therefore define a *summary-based reward*, computed as the cross-entropy loss of an open-source solver. Given a query $q$ and answer $a = (a_1, \ldots, a_{|a|})$ from chunk $x$, the reward is:

$$R_{\text{qual}} = \frac{1}{1 + \frac{1}{|a|} \sum_{i=1}^{|a|} \log P\big(a_i \mid q, x_C, a_{<i}\big)}, \tag{2}$$

where $|a|$ is the answer length and $a_{<i} = (a_1, \ldots, a_{i-1})$ its prefix. Intuitively, the reward increases when $x_C$ enables the solver to predict the answer with high probability, indicating that the compressed input has preserved sufficient task-relevant information.

**Dual-objective reward.** To balance compactness and informativeness, we combine both objectives into a single reward signal (Equation 3). This combined reward simultaneously penalizes the unnecessary verbosity, avoids harmful over compression, and promotes representations that are both efficient and effective for task performance.

$$R = R_{\text{qual}} + R_{\text{len}}. \tag{3}$$

**Training.** Further details of GRPO training are given in Sec. C.1. While we also experimented with DPO and ORPO, these approaches did not outperform the GRPO-based pipeline; we describe these setups in Sec. C.2 of the Appendix.

## 4.3 EVALUATION

To assess the effectiveness of our approach, we evaluate model performance under two complementary task settings: *summarization* and *QA*. Our primary focus is (i) to examine whether the compressed representations are **question-agnostic**, i.e., they preserve general information that supports diverse queries, and (ii) to verify whether they retain sufficient **abstraction** to follow the intent of summarization tasks. Although the model is trained exclusively on the `MeetingBank` dataset, we evaluate it across multiple out-of-distribution (OOD) benchmarks, including `GSM8K` for QA and `LongBench` for both summarization and QA. This setup tests generalization across tasks with differing requirements, ranging from factual short reasoning to long-document summarization. The evaluation pipeline operates as follows: a compressed text $x_C$ is first generated, which is then fed into the solver model to either produce a summary or answer task-specific questions.

We report the evaluation results in Table 8. For summarization tasks, we adopt the Bert-F1 score as the primary metric, while for QA tasks we use accuracy (i.e 1 if the predicted answer matches the ground truth, and 0 otherwise). For `LongBench`, which includes multiple subtasks within the same dataset, we aggregate results by averaging over all QA-related tasks (e.g., single-document QA and few-shot QA) and separately averaging over all summarization-related tasks. Full, fine-grained results for each subtask are provided in the Appendix D.

Table 2: Question quality and summary similarity under different compression ratios (CR). $\Delta_{CR}$ is the absolute deviation from the target compression ratio (lower is better).

| CR | Models | Question Score | | | | | | Summary Similarity | | | |
|---|---|---|---|---|---|---|---|---|---|---|---|
| | | GSM8K | | LongBench | | MeetingBank | | MeetingBank | | LongBench | |
| | | $\Delta_{CR}$ | EM | $\Delta_{CR}$ | Accuracy | $\Delta_{CR}$ | Accuracy | $\Delta_{CR}$ | BERT-F1 | $\Delta_{CR}$ | BERT-F1 |
| | Lingua 2 | 0.04 | 0.67 | -0.01 | 16.67 | -0.01 | 16.70 | -0.01 | 0.8641 | -0.01 | 0.8564 |
| | gpt-4.1-mini | 0.24 | 39.33 | 0.12 | 20.00 | 0.07 | 18.27 | 0.07 | 0.8865 | 0.09 | 0.8766 |
| | TG-gpt-4.1-mini | 0.05 | 27.33 | 0.11 | 20.82 | 0.07 | 25.05 | 0.07 | 0.8874 | 0.08 | 0.8796 |
| | Qwen3-4b | 0.42 | 63.00 | 0.09 | 20.00 | 0.06 | 14.89 | 0.06 | 0.8629 | 0.10 | 0.8622 |
| 0.1 | TG-Qwen3-4b | 0.35 | 50.67 | 0.13 | 20.22 | 0.16 | 23.02 | 0.16 | 0.8885 | 0.15 | 0.8787 |
| | SFT | 0.30 | 54.33 | 0.07 | 17.41 | 0.07 | 17.96 | 0.07 | 0.8841 | 0.06 | 0.8754 |
| | TG-SFT | 0.14 | 27.33 | 0.04 | 19.71 | 0.03 | 15.30 | 0.03 | 0.8804 | 0.03 | 0.8718 |
| | GRPO | 0.27 | 7.33 | 0.00 | 17.63 | 0.02 | 17.26 | 0.02 | 0.8461 | -0.01 | 0.8402 |
| | CMPRSR | **0.00** | **12.67** | **-0.02** | **19.67** | **-0.01** | **20.05** | **-0.01** | **0.8773** | **-0.02** | **0.8691** |
| | Lingua 2 | 0.08 | 12.33 | -0.01 | 19.74 | -0.03 | 32.70 | -0.03 | 0.8913 | -0.02 | 0.8773 |
| | gpt-4.1-mini | 0.24 | 82.00 | 0.13 | 19.00 | 0.00 | 27.61 | 0.00 | 0.8989 | 0.07 | 0.8829 |
| | TG-gpt-4.1-mini | 0.16 | 66.33 | 0.18 | 20.70 | 0.07 | 36.70 | 0.07 | 0.9014 | 0.16 | 0.8886 |
| | Qwen3-4b | 0.32 | 75.00 | -0.01 | 20.00 | -0.07 | 18.69 | -0.07 | 0.8823 | -0.03 | 0.8713 |
| 0.3 | TG-Qwen3-4b | 0.30 | 69.67 | 0.06 | 20.39 | 0.05 | 29.62 | 0.05 | 0.8978 | 0.07 | 0.8822 |
| | SFT | 0.27 | 78.00 | 0.09 | 21.77 | 0.02 | 26.38 | 0.02 | 0.8971 | 0.03 | 0.8817 |
| | TG-SFT | 0.17 | 71.00 | 0.06 | 20.10 | 0.00 | 26.23 | 0.00 | 0.8962 | 0.00 | 0.8803 |
| | GRPO | 0.12 | 13.33 | 0.06 | 18.80 | -0.01 | 25.40 | -0.01 | 0.8641 | -0.01 | 0.8603 |
| | CMPRSR | **0.00** | **35.33** | **-0.01** | **20.10** | **-0.01** | **33.20** | **-0.01** | **0.9009** | **-0.04** | **0.8835** |
| | Lingua 2 | 0.09 | 49.33 | -0.01 | 17.14 | -0.03 | 41.14 | -0.03 | 0.9020 | -0.02 | 0.8893 |
| | gpt-4.1-mini | 0.14 | 87.00 | **-0.01** | **21.10** | -0.17 | 28.36 | -0.17 | 0.8975 | -0.07 | 0.8857 |
| | TG-gpt-4.1-mini | 0.14 | 82.67 | 0.09 | 19.67 | -0.02 | 39.90 | -0.02 | 0.9039 | 0.12 | 0.8913 |
| | Qwen3-4b | 0.12 | 76.67 | -0.16 | 19.18 | -0.24 | 20.22 | -0.24 | 0.8868 | -0.19 | 0.8748 |
| 0.5 | TG-Qwen3-4b | 0.24 | 74.67 | -0.06 | 19.76 | -0.06 | 32.78 | -0.06 | 0.9004 | -0.04 | 0.8858 |
| | SFT | 0.15 | 83.33 | -0.01 | 20.72 | -0.10 | 30.87 | -0.10 | 0.8965 | -0.08 | 0.8852 |
| | TG-SFT | 0.06 | 81.67 | 0.00 | 20.83 | -0.10 | 30.81 | -0.10 | 0.9018 | -0.09 | 0.8837 |
| | GRPO | -0.05 | 15.00 | -0.01 | 19.57 | -0.11 | 28.95 | -0.11 | 0.8704 | -0.09 | 0.8629 |
| | CMPRSR | **-0.06** | **62.33** | 0.03 | 20.20 | **-0.01** | **41.95** | **-0.01** | **0.9083** | **-0.04** | **0.8917** |

Based on Table 8, higher raw scores are often achieved at the cost of exceeding the target CR, i.e positive $\Delta_{CR}$. Since our objective is to maximize task performance *under strict constraint of* $\Delta_{CR} \leq 0$, we do not select the highest score. Instead, we focus on models that respect the CR budget while still delivering competitive outcomes. Under this criterion, SFT and GRPO occasionally yield strong performance, but only when tolerating higher $\Delta_{CR}$ values, i.e., compressing less than required. To improve on this, we explore GRPO and also `TextGrad` as cost-efficient alternatives. For the vanilla model, `TextGrad` works well with GPT-4.1 but not with Qwen; in contrast, it yields better results when combined with SFT on Qwen. However, our proposed `CMPRSR` model consistently achieves the best trade-off across both QA and summarization tasks. By effectively combining SFT with GRPO, it maintains $\Delta_{CR} <= 0$ , ensuring strict adherence to the target CR while outperforming baseline models. Figure 4 further investigates model adherence to the CR across training stages:

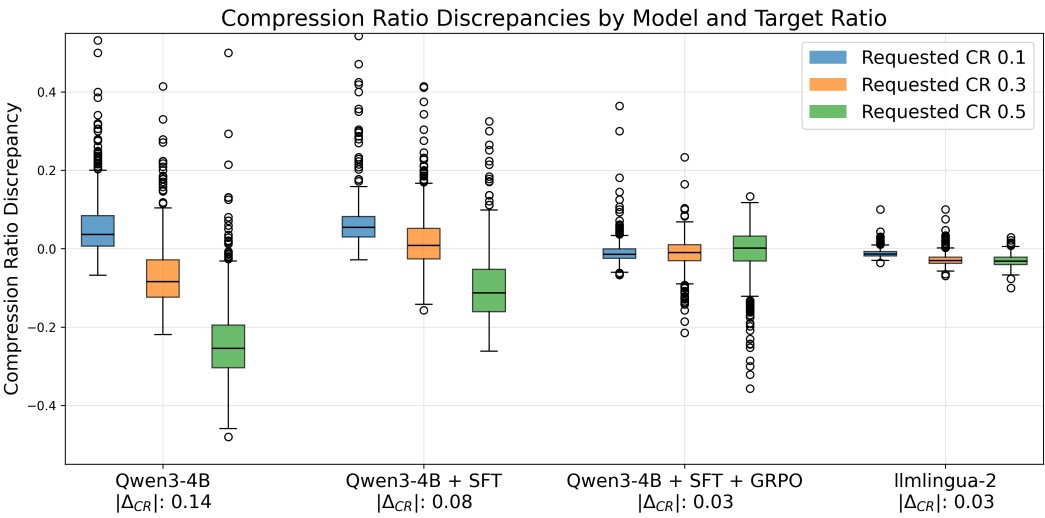

Figure 4: $\Delta_{CR}$ : for different models.

*LLMLingua-2* demonstrates stronger adherence due to its inherently extractive compression strategy. In contrast, the Qwen3-4B model fails to match the requested CR, often producing outputs with large negative $\Delta_{CR}$ values. This suggests that, without targeted training, the model tends to over-compress. SFT partially mitigates this issue by reducing the average discrepancy, though it still leaves considerable variance. However, the CMPRSR model, which combines SFT with GRPO, achieves the most balanced results. It keeps discrepancies close to zero while simultaneously reducing variance across all CR. This progression highlights a clear evolution of target adherence throughout the training pipeline: from poor alignment in the vanilla model, to partial improvement with single-objective fine-tuning, to robust adherence with CMPRSR. Moreover, the reduction in variance shows that CMPRSR is not only accurate on average but also reliable at the instance level.

## 4.4 PRACTICAL APPLICABILITY DISCUSSION

While **Cmprsr** outperforms *LLMLingua-2* across the compression rate spectrum, it is important to note that (i) it is based on a **4B** backbone, while *LLMLingua-2* is a fine-tuned *xlm-roberta-large* of **0.55B** parameters (ii) the autoregressive nature of **Cmprsr** implies latency overhead. This, however, is unlikely to prevent widespread adoption of LLM-based compressors. First, most of the question-agnostic use-cases discussed in Sec. 1 allow for the the usage of the pre-computed compressions, which decreases the importance of latency. Secondly, the usage of either **0.55B** or a **4B** model will come at a fraction of the savings, generated by optimized inference of the *Target* model. Quantitatively, LLama-3.2-3B inference cost is \$0.06 (both input and output at together.ai) and GPT-5 is \$1.25 per $10^6$ input tokens, meaning that in case of $CR = 0.3$, the input-incurred GPT-5 cost would be reduced by $64\%$, while in the limit of the free compression the savings would amount to $70\%$.

### 4.5 ANALYSIS

**Cross-entropy under summarization.** Table 5 reports cross-entropy (CE) loss for the summarization task across compression rates. CMPRSR yields consistently lower CE than LLMLINGUA-2 throughout the range $CR \in [0.1, 0.7]$. Since both compressors were trained with target rates in this interval, we restrict evaluation accordingly. Notably, CE remains near its uncompressed baseline even at $CR = 0.5$, indicating that substantial prompt reduction can be achieved without materially degrading next-token predictive quality. We observe the same trend for GEMMA-3 models of multiple sizes (see Sec. D.4 in Appendix), suggesting that **Cmprsr** generalizes across architectures.

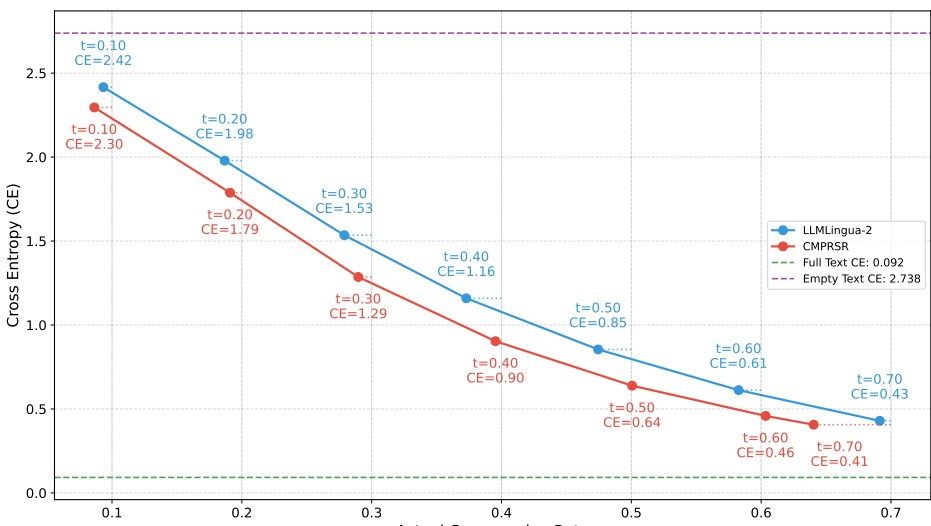

Figure 5: Cross-Entropy (CE) losses for the summarization task under two prompt compression methods, LLMLingua-2 and Cmprsr, across target compression rates. Lower CE indicates better information preservation. The dashed green line marks the mean CE without compression, while the dashed purple line marks CE with empty compression. Dotted horizontal connectors link each target to the actually achieved rate at the same CE, so shorter connectors imply better rate control. Numeric tags denote the corresponding target $t$ and $CE$. Results are averaged over 1000 Meeting-Bank validation samples using the Llama-3.3-70B 4-bit quantized model.

**Local structure preservation.** To probe how compression affects surface form, we compute $n$-gram overlaps between compressed and original prompts, (Table 11). **Cmprsr** outputs show higher 2-gram and 3-gram overlaps, especially at lower CR. **Cmprsr** preserves local phrase structure more faithfully, whereas LLMLINGUA-2 exhibits higher fragmentation. This pattern supports the hypothesis that **Cmprsr** maintains semantic adequacy and short-range syntactic cohesion, which can be advantageous for downstream components that rely on multi-token dependencies.

## 5 CONCLUSION

We propose using LLMs for abstractive prompt compression, starting with a comprehensive benchmark of 25 off-the-shelf models. Upon that, we improve the best vanilla compressor—*GPT-4.1-mini*—with the *Textgrad*-based meta-prompt optimization, and one of the smaller models—*Qwen-3-4B*—using *SFT* and *GRPO* posttraining. The resulting **Cmprsr** outperforms both the *extractive* SOTA *LLMLingua-2* and the leading abstractive vanilla – Textgrad-boosted *gpt-4.1-mini*.

We argue that "small LLMs" have become capable enough for the practitioners to shift their focus to the paradigm of smaller LLMs compressing prompts for the larger ones. However, vanilla models do not exhibit sufficient performance and should be post-trained for this specific task. **Cmprsr** both pioneers this line of research and generates SOTA level compressions, suggesting immediate practical value for the community.

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

# A COMPRESSIONBENCH DETAILS

In this section, we detail the exact versions of the models we use (Table 3), the full Compression-Bench results on MB (Table 4), and on GSM8k (Table 5).

## A.1 INITIAL COMPRESSOR SYSTEM PROMPT.

The following prompt was used as the initial system prompt in iteration 0 of TEXTGRAD. Additionally, it was used in most experiments and benchmarking across GSM8k, LongBench, and MeetingBank, unless stated otherwise. This prompt was explicitly chosen to be dataset-agnostic and work with most downstream applications.

```
You are an agent whose task is to compress prompts passed to you by the user.  Preserve only the
necessary information, relationships, and required answer format.  Remove all unnecessary details,
drop redundant information, and rephrase what can be rephrased without information loss.  You can
aggressively shorten words, drop prepositions and articles – do whatever it takes to shorten the prompt
to the absolute bare minimum, while avoiding the loss of any important information contained in the
prompt.  Use compact notation, single-letter variables, and standard abbreviations.  Clarify ambiguities
minimally.  Stop compressing if it risks clarity or solvability.  If the prompt is posed as a question,
you must output the compressed question, keeping the question format.  Do not answer the question; only
compress it.
```

## A.2 MODELS

We present all models using their short names, while ensuring that the corresponding full model names are introduced when first mentioned.

Table 3: Models and their short names

| Full name | Short name |
|---|---|
| gpt-5-nano-2025-08-07 | gpt-5-nano |
| gpt-4.1-mini-2025-04-14 | gpt-4.1-mini |
| gpt-5-mini-2025-08-07 | gpt-5-mini |
| gpt-5-2025-08-07 | gpt-5 |
| gemini-2.0-flash-lite | gemini-2.0-flash-lite |
| gpt-4.1-nano-2025-04-14 | gpt-4.1-nano |
| gpt-4.1-2025-04-14 | gpt-4.1 |
| gemini-2.5-flash (1500 tkns reasoning) | gemini-2.5-flash |
| o4-mini-2025-04-16 (low reasoning) | o4-mini-2025-04-16 |
| google/gemma-3-12b-it | gemma-3-12b-it |
| mistralai/Mistral-Small-3.1-24B-Instruct-2503 | Mistral-Small-3.1-24B |
| deepseek-ai/DeepSeek-V3 | DeepSeek-V3 |
| meta-llama/Llama-3.3-70B-Instruct | Llama-3.3-70B |
| Qwen/Qwen3-30B-A3B-Instruct-2507 | Qwen3-30B-A3B |
| Qwen/Qwen3-235B-A22B-Instruct-2507 | Qwen3-235B-A22B |
| Qwen/Qwen2.5-32B-Instruct | Qwen2.5-32B |
| google/gemma-3-27b-it | gemma-3-27b |
| meta-llama/Meta-Llama-3.1-405B-Instruct | Meta-Llama-3.1-405B |
| Qwen/Qwen2.5-14B-Instruct | Qwen2.5-14B |
| meta-llama/Llama-3.2-3B-Instruct | Llama-3.2-3B |
| Qwen/Qwen3-4B-Instruct-2507 | Qwen3-4B |
| Qwen/Qwen2.5-7B-Instruct | Qwen2.5-7B |
| google/gemma-3-4b-it | gemma-3-4b |
| meta-llama/Llama-3.1-8B-Instruct | Llama-3.1-8B |
| Qwen/Qwen2.5-3B-Instruct | Qwen2.5-3B |
| google/flan-t5-xxl | flan-t5-xxl |
| google/flan-t5-xl | flan-t5-xl |

## A.3 COMPRESSIONBENCH RESULTS ON MEETINGBANK

We present the full results for all models used to benchmark the LLM as a compressor. Our evaluation includes 9 closed-source models and 18 open-source models spanning a wide range of parameter scales and architectural families. All experiments were conducted on the validation split of the MeetingBank dataset, which provides diverse and realistic meeting transcripts for evaluation. From this split, we randomly selected 100 samples, each of which was divided into multiple chunks when exceeding the 512-token limit (the exact number of chunks varies depending on the tokenizer used).

Table 4: Compression performance of various vanilla models on the MeetingBank transcripts.

| ratio | $\Delta_{CR}$ | | | BERT-F1 | | | QA | | |
|---|---|---|---|---|---|---|---|---|---|
| | 0.1 | 0.3 | 0.5 | 0.1 | 0.3 | 0.5 | 0.1 | 0.3 | 0.5 |
| **Closed-source models** | | | | | | | | | |
| gpt-5-nano | 0.19 | 0.17 | 0.04 | 0.87 | 0.88 | 0.88 | 0.25 | 0.30 | 0.31 |
| gpt-4.1-mini | 0.07 | -0.00 | -0.17 | 0.89 | 0.90 | 0.90 | 0.20 | 0.29 | 0.30 |
| gpt-5-mini | 0.13 | 0.11 | 0.10 | 0.86 | 0.87 | 0.87 | 0.19 | 0.25 | 0.27 |
| gpt-5 | 0.10 | 0.08 | 0.06 | 0.87 | 0.88 | 0.88 | 0.18 | 0.25 | 0.25 |
| gemini-2.0-flash-lite | 0.09 | 0.00 | -0.17 | 0.88 | 0.89 | 0.89 | 0.17 | 0.24 | 0.25 |
| gpt-4.1-nano | 0.04 | -0.02 | -0.19 | 0.87 | 0.89 | 0.89 | 0.15 | 0.26 | 0.25 |
| gpt-4.1 | 0.06 | -0.02 | -0.19 | 0.88 | 0.89 | 0.89 | 0.17 | 0.23 | 0.25 |
| gemini-2.5-flash | 0.03 | -0.03 | -0.16 | 0.88 | 0.89 | 0.89 | 0.15 | 0.22 | 0.24 |
| o4-mini | 0.10 | 0.05 | -0.17 | 0.88 | 0.88 | 0.88 | 0.14 | 0.22 | 0.21 |
| **Large Open-source models ($> 10B$)** | | | | | | | | | |
| gemma-3-12b-it | 0.18 | 0.07 | -0.05 | 0.89 | 0.89 | 0.89 | 0.22 | 0.25 | 0.26 |
| Mistral-Small-3.1-24B | 0.14 | -0.04 | -0.23 | 0.88 | 0.89 | 0.89 | 0.22 | 0.24 | 0.23 |
| DeepSeek-V3 | 0.11 | -0.05 | -0.24 | 0.89 | 0.89 | 0.89 | 0.21 | 0.24 | 0.25 |
| Llama-3.3-70B | 0.07 | 0.15 | -0.02 | 0.88 | 0.88 | 0.88 | 0.20 | 0.24 | 0.24 |
| Qwen3-30B-A3B | 0.15 | 0.01 | -0.17 | 0.88 | 0.89 | 0.89 | 0.20 | 0.23 | 0.23 |
| Qwen3-235B-A22B | 0.13 | 0.03 | -0.11 | 0.88 | 0.89 | 0.89 | 0.18 | 0.21 | 0.23 |
| Qwen2.5-32B | 0.08 | -0.06 | -0.25 | 0.88 | 0.88 | 0.88 | 0.18 | 0.21 | 0.22 |
| gemma-3-27b | 0.16 | 0.13 | 0.21 | 0.88 | 0.88 | 0.88 | 0.16 | 0.18 | 0.20 |
| Meta-Llama-3.1-405B | 0.04 | -0.10 | -0.30 | 0.86 | 0.87 | 0.87 | 0.18 | 0.17 | 0.18 |
| Qwen2.5-14B | 0.11 | -0.03 | -0.21 | 0.87 | 0.88 | 0.87 | 0.15 | 0.17 | 0.17 |
| **Small Open-source models ($< 10B$)** | | | | | | | | | |
| Llama-3.2-3B | 0.05 | -0.10 | -0.30 | 0.87 | 0.87 | 0.87 | 0.17 | 0.21 | 0.22 |
| Qwen3-4B | 0.05 | -0.08 | -0.26 | 0.86 | 0.88 | 0.88 | 0.16 | 0.21 | 0.22 |
| Qwen2.5-7B | 0.08 | -0.11 | -0.31 | 0.87 | 0.88 | 0.88 | 0.16 | 0.19 | 0.18 |
| gemma-3-4b | 0.00 | -0.15 | -0.35 | 0.87 | 0.87 | 0.88 | 0.13 | 0.17 | 0.19 |
| Llama-3.1-8B | -0.02 | -0.19 | -0.39 | 0.85 | 0.86 | 0.86 | 0.14 | 0.17 | 0.15 |
| Qwen2.5-3B | 0.09 | -0.08 | -0.24 | 0.85 | 0.86 | 0.86 | 0.10 | 0.12 | 0.12 |
| **Encoder-Decoder Models** | | | | | | | | | |
| flan-t5-xxl | 0.60 | 0.39 | 0.18 | 0.89 | 0.89 | 0.89 | 0.31 | 0.30 | 0.31 |
| flan-t5-xl | 0.16 | -0.04 | -0.24 | 0.87 | 0.87 | 0.87 | 0.17 | 0.17 | 0.16 |
| **Extractive** | | | | | | | | | |
| llmlingua-2 | -0.01 | -0.03 | -0.03 | 0.86 | 0.89 | 0.9 | 0.16 | 0.34 | 0.42 |

A.4 COMPRESSIONBENCH RESULTS ON GSM8K

We also report results using the GSM8k dataset to further evaluate the models under a different task setting. In this case, all input sequences are shorter than 512 tokens, so no chunking was required during preprocessing. We randomly selected 300 samples from the validation split of the dataset, ensuring sufficient coverage to assess the performance of both open- and closed-source models under this setting. In the math dataset, where the context is very short, LLM-Lingua performs poorly, which is expected since extractive models cannot effectively compress mathematical problem statements without abstraction, rephrasing, or reordering of words. Meanwhile, the decoder-only vanilla model shows relatively strong performance but does not adhere to the target compression ratio.

Table 5: Compression performance of various vanilla models on the GSM8k problems.

| | $\Delta_{CR}$ | | | Accuracy | | |
|---|---|---|---|---|---|---|
| Requested CR | 0.1 | 0.3 | 0.5 | 0.1 | 0.3 | 0.5 |
| **Closed-source models** | | | | | | |
| gpt-5-nano | 0.41 | 0.44 | 0.25 | 0.69 | 0.82 | 0.87 |
| gpt-5 | 0.23 | 0.22 | 0.13 | 0.61 | 0.86 | 0.90 |
| gpt-5-mini | 0.20 | 0.23 | 0.12 | 0.55 | 0.85 | 0.88 |
| gpt-4.1 | 0.22 | 0.17 | 0.05 | 0.58 | 0.82 | 0.86 |
| gemini-2.5-flash | 1.27 | 0.17 | 0.17 | 0.52 | 0.79 | 0.87 |
| gemini-2.0-flash-lite | 0.34 | 0.31 | 0.16 | 0.55 | 0.78 | 0.80 |
| gpt-4.1-mini | 0.20 | 0.17 | 0.06 | 0.43 | 0.80 | 0.85 |
| gpt-4.1-nano-2025-04-14 | 0.10 | 0.13 | 0.01 | 0.17 | 0.62 | 0.76 |
| **Large Open-source models ($> 10B$)** | | | | | | |
| Meta-Llama-3.1-405B | 0.40 | 0.33 | 0.13 | 0.78 | 0.84 | 0.84 |
| Qwen3-235B-A22B | 0.36 | 0.21 | 0.04 | 0.77 | 0.81 | 0.84 |
| Qwen2.5-32B | 0.40 | 0.24 | 0.06 | 0.77 | 0.80 | 0.83 |
| DeepSeek-V3 | 0.34 | 0.21 | 0.04 | 0.70 | 0.80 | 0.83 |
| Qwen3-30B-A3B | 0.36 | 0.21 | 0.04 | 0.69 | 0.77 | 0.80 |
| gemma-3-27b-it | 0.33 | 0.32 | 0.45 | 0.66 | 0.74 | 0.80 |
| Mistral-Small-3.1-24B | 0.41 | 0.28 | 0.10 | 0.67 | 0.75 | 0.77 |
| Llama-3.3-70B | 0.25 | 0.22 | 0.06 | 0.51 | 0.73 | 0.81 |
| gemma-3-12b-it | 0.29 | 0.28 | 0.13 | 0.45 | 0.74 | 0.74 |
| Qwen2.5-14B | 0.29 | 0.20 | 0.02 | 0.49 | 0.70 | 0.70 |
| **Small Open-source models ($< 10B$)** | | | | | | |
| Qwen3-4B-Instruct | 0.37 | 0.22 | 0.05 | 0.70 | 0.74 | 0.78 |
| Llama-3.1-8B | 0.36 | 0.23 | 0.05 | 0.55 | 0.62 | 0.63 |
| Qwen2.5-7B | 0.27 | 0.14 | -0.04 | 0.50 | 0.57 | 0.64 |
| gemma-3-4b | 0.15 | 0.13 | -0.01 | 0.15 | 0.45 | 0.58 |
| Llama-3.2-3B | 0.25 | 0.19 | 0.02 | 0.24 | 0.37 | 0.39 |
| Qwen2.5-3B | 0.08 | 0.12 | -0.11 | 0.15 | 0.21 | 0.28 |
| **Encoder-Decoder Models** | | | | | | |
| llmlingua-2 | 0.02 | 0.02 | -0.00 | 0.01 | 0.19 | 0.49 |

## B TEXTGRAD IMPLEMENTATION DETAILS

We utilize TEXTGRAD to optimize the compressor's *system prompt* using question answering (QA) on the MeetingBank dataset Hu et al. (2023) as the source of the learning signal. In short, our method runs the following loop:

1. Run the end-to-end pipeline on a batch of MeetingBank transcripts and score outputs against the ground truth;

2. Ask an optimizer-LLM to analyze errors in the output and generate improvement suggestions;

3. Propagate this feedback to the upstream nodes and propose a revised prompt;

4. Accept the updated prompt if it outperforms the initial prompt on a hold-out set;

5. Re-evaluate the pipeline with the (possibly) updated prompt and repeat until reaching the budget constraints.

Figure 6 illustrates the computation graph involved in our prompt optimization procedure. The grey blocks represent text-based variables, the blue color denotes LLM nodes, and the optimized compressor system prompt variable is depicted in indigo. Meanwhile, the purple blocks show excerpts from TEXTGRAD textual gradients. In this example, the natural language feedback generated by the optimizer LLM (i) identifies a question answering mistake (namely, the Answering LLM predicts "Councilor Bark" instead of the correct "Councilor Bok") and (ii) instructs the compressor to explicitly preserve named entities to prevent such mistakes in the future.

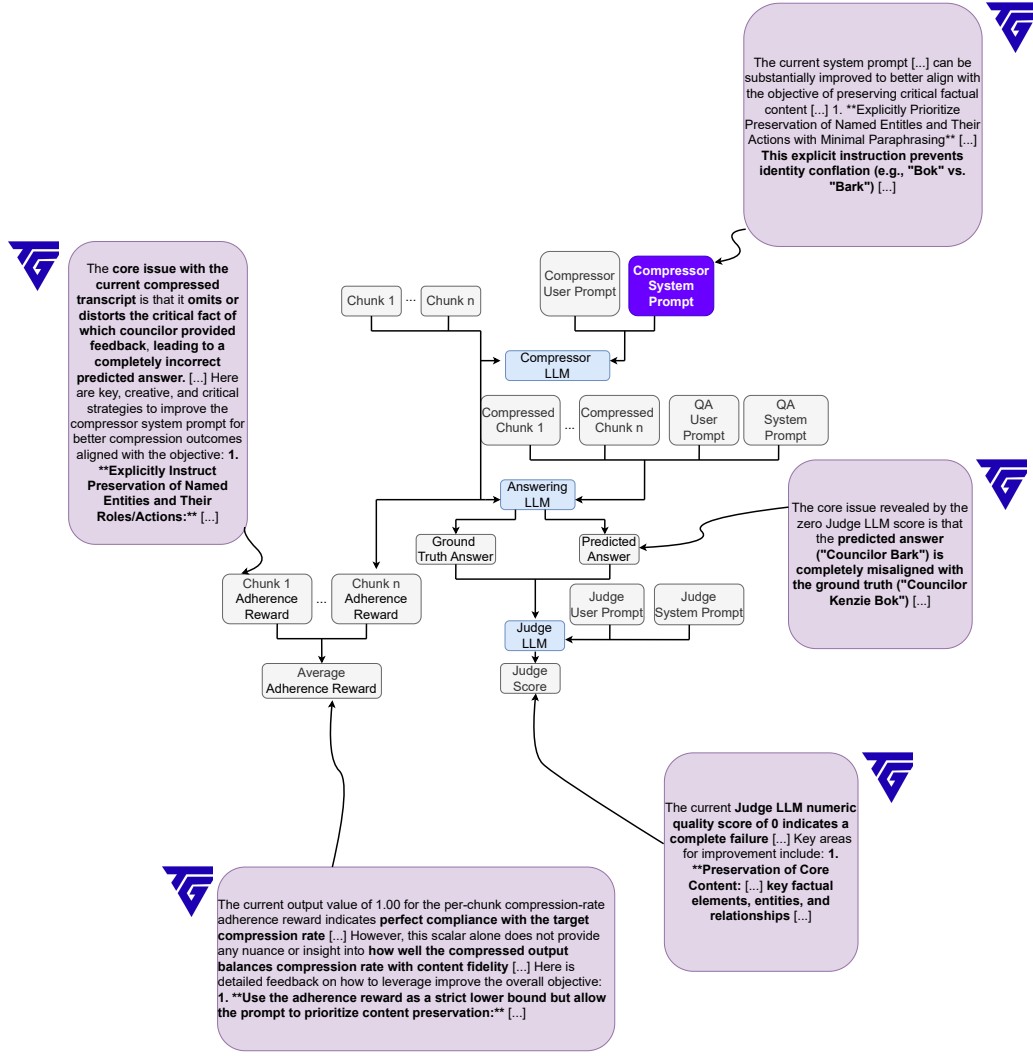

Figure 6: **TEXTGRAD Computation Graph on the MeetingBank QA dataset**.

The main goal of prompt optimization is to improve (i) downstream QA quality with compressed context, and (ii) adherence to a user-specified compression rate (CR).

**QA quality scoring.** To assess information retention, we use synthetic QA pairs generated from the uncompressed MeetingBank transcript. Given a compressed transcript as context, the *answering LLM* (gpt-4.1-mini) predicts an answer. Next, the *judge LLM* (gpt-4.1-mini) scores this prediction against the ground truth answer on a discrete 0 to 10 scale.

**CR Adherence scoring.** For a chunk with the target compression rate $r_{\text{tgt}}$ and the actual produced compression rate $r_{\text{act}} = \frac{N_{\text{comp}}}{N_{\text{orig}}}$, the per–chunk adherence reward $\text{Adh}_C$ is given by:

$$\text{Adh}_C = 1 - \max\big(0,\, r_{\text{act}} - r_{\text{tgt}}\big), \qquad \text{Adh}_C \in (-\infty,\, 1], \qquad \text{higher is better} \tag{4}$$

Note that in the above formula $N_{\text{comp}}$ is the number of tokens in the produced compression and $N_{\text{orig}}$ is the number of tokens in the original input.

For a transcript $T$ split into $K$ chunks $\{C_k\}_{k=1}^{K}$, we report the transcript–level CR adherence reward as the average of the rewards for each of its constituent chunks:

$$\text{Adh}_T = \frac{1}{K} \sum_{k=1}^{K} \text{Adh}_{C_k} \tag{5}$$

**Pseudocode overview.** We can now examine the TEXTGRAD prompt optimization approach in greater detail. Overall, we can identify three main components of the algorithm.

## 1. Sampling and Compression

1. Sample a small batch of $TextGradBatchSize = 2$ training transcripts.

2. Split each transcript into chunks of at most $MaxChunkTokens = 512$ tokens.

3. For each chunk, draw a target compression rate $r_{\text{tgt}} \sim \mathcal{U}(0.1,\, 0.7)$ and compute the token budget $N_{\text{tgt}}$ as follows:
$$N_{\text{tgt}} = \big\lfloor r_{\text{tgt}} \cdot N_{\text{orig}} \big\rfloor$$

4. Prompt the LLM-based compressor to produce a compressed chunk under budget $N_{\text{tgt}}$.

5. Concatenate the compressed chunks to obtain the compressed transcript.

## 2. QA-based Evaluation

1. For each compressed transcript $t$, evaluate $Q = 20$ synthetic questions and retain the $NumHardQ = 2$ questions with the lowest judge scores (i.e., the hardest questions). Let $H_t \subseteq \{1, \ldots, Q\}$ be the indices of these challenging questions ($|H_t| = NumHardQ$). The per-transcript hard-question judge average is then:
$$\text{JudgeAvg}_t = \frac{1}{NumHardQ} \sum_{i \in H_t} \text{JudgeScore}_{t,i}$$

2. Aggregate per-batch metrics over $TextGradBatchSize$ transcripts. If transcript $t$ is split into $C_t$ chunks with per-chunk adherence $\text{Adh}_{t,j}$, $j = 1, \ldots, C_t$, we define the per-transcript adherence as:
$$\text{Adh}_t = \frac{1}{C_t} \sum_{j=1}^{C_t} \text{Adh}_{t,j}$$

The per-batch aggregates are then:

$$\text{JudgeAvg} = \frac{1}{B} \sum_{t=1}^{B} \text{JudgeAvg}_t, \qquad \text{AdhAvg} = \frac{1}{B} \sum_{t=1}^{B} \text{Adh}_{T_t}$$

### 3. TEXTGRAD Optimization Step

1. Use $JudgeAvg$ and $AdhAvg$ as TEXTGRAD optimization objectives to generate a new candidate compressor's system prompt.

2. Validate the candidate on 100 hold-out transcripts.

3. Accept the update if the new prompt increases either $JudgeAvg$ or $AdhAvg$ when compared to the previous prompt.

Combining the three key components of our prompt optimization approach, we present the complete algorithm in pseudocode section below.

---

**Algorithm 1** TextGrad Prompt Optimization via QA and Compression-Rate Adherence

---

**Inputs:** $InitialPrompt \leftarrow$ original system prompt for the compressor;
1: $\mathcal{V}$: validation set of 100 MeetingBank transcripts;
2: $\{\mathcal{B}_1, \ldots, \mathcal{B}_m\}$: training batches (size $= TextgradBatchSize = 2$);
3: $SyntheticQAs$: mapping from transcript id $\rightarrow$ the list of question-answer tuples $(q, a)$;
4: $AnswerLLM$: model used to answer questions (gpt-4.1-mini);
5: $CompressorLLM$: abstractive compressor (its system prompt is optimized, Qwen3-4B);
6: $JudgeLLM$: LLM judge scoring predicted vs. ground truth answer on the scale $[0, 10]$ (gpt-4.1-mini);
7: Hyperparameters: $MaxChunkTokens = 512$, $NumHardQ = 2$, target ratio range $[0.1, 0.7]$.
**Outputs:** OptimizedPrompt
8: **procedure** OPTIMIZEPROMPT
9: $\quad CurrentPrompt \leftarrow InitialPrompt$
10: $\quad (PrevJudgeAvg, PrevAdherenceAvg) \leftarrow$ MEASUREPERFORMANCE$(\mathcal{V}, CurrentPrompt)$ $\quad\quad\triangleright$ average judge score and average rate-adherence
11: $\quad$ **for** each batch $\mathcal{B}$ in $\{\mathcal{B}_1, \ldots, \mathcal{B}_m\}$ **do**
12: $\quad\quad TrainingPoints \leftarrow \emptyset$
13: $\quad\quad$ **for** each transcript $T$ in $\mathcal{B}$ **do**
14: $\quad\quad\quad (CompressedT, Adh_T) \leftarrow$ COMPRESSTRANSCRIPT$(T, CompressorLLM, CurrentPrompt, MaxChunkTokens)$
15: $\quad\quad\quad QAResults \leftarrow \emptyset$
16: $\quad\quad\quad$ **for** each $(q, a)$ in $SyntheticQAs[T.\text{id}]$ **do**
17: $\quad\quad\quad\quad pred \leftarrow AnswerLLM(q, \text{context} = CompressedT)$
18: $\quad\quad\quad\quad score \leftarrow JudgeLLM(pred, q, a)$ $\quad\quad\quad\quad\quad\quad\quad\triangleright 0 \leq score \leq 10$
19: $\quad\quad\quad\quad QAResults \leftarrow QAResults \cup \{\langle q, a, pred, score \rangle\}$
20: $\quad\quad\quad$ **end for**
21: $\quad\quad\quad Hardest \leftarrow$ BOTTOMK$(QAResults, k = NumHardQ, \text{by } score)$ $\quad\quad\triangleright$ lowest judge scores
22: $\quad\quad\quad$ **for** each $r \in Hardest$ **do**
23: $\quad\quad\quad\quad TrainingPoints \leftarrow TrainingPoints \cup \{\langle r.q, r.a, r.pred, r.score, Adh_T, T.\text{id}\rangle\}$
24: $\quad\quad\quad$ **end for**
25: $\quad\quad$ **end for**
26: $\quad\quad CandidatePrompt \leftarrow$ TEXTGRADBACKWARD$(TrainingPoints, CurrentPrompt)$
27: $\quad\quad (NewJudgeAvg, NewAdherenceAvg) \leftarrow$ MEASUREPERFORMANCE$(\mathcal{V}, CandidatePrompt)$
28: $\quad\quad$ **if** $(NewJudgeAvg > PrevJudgeAvg) \lor (NewAdherenceAvg > PrevAdherenceAvg)$ **then**
29: $\quad\quad\quad CurrentPrompt \leftarrow CandidatePrompt$
30: $\quad\quad\quad (PrevJudgeAvg, PrevAdherenceAvg) \leftarrow (NewJudgeAvg, NewAdherenceAvg)$
31: $\quad\quad$ **end if**
32: $\quad$ **end for**
33: $\quad$ **return** $CurrentPrompt$ $\quad\quad\quad\quad\quad\quad\quad\quad\quad\quad\quad\quad\quad\triangleright$ OPTIMIZEDPROMPT
34: **end procedure**

---

**Notable prompts.** We will now present selected prompts used in the TEXTGRAD pipeline.

**Compressor User Prompt.** Following TEXTGRAD's practice of reusing a shared system prompt across inputs, we specify the desired token budget in the user prompt. The user prompt is lean by design, as most of the compression instructions will be supplied in the system prompt.

```
Please compress the text below.  The length of the resulting compression must be
{desired_length} tokens.
Text to compress:
```

**Best Quality System Prompt.** The best-quality TEXTGRAD prompt aggregates optimizer LLM strategies over multiple TEXTGRAD iterations. It explicitly stresses the importance of named entities, numeric values, and other facts that are likely to appear in the downstream synthetic questions. The updated prompt also uses few-shot demonstrations to capture the lessons learned from mistakes on difficult questions. Through numerous examples and instructions, the updated prompt distills part of the optimizer LLM's knowledge into the student Qwen3-4B compressor.

```
You are an agent tasked with compressing transcript chunks to preserve all information necessary for
accurate question answering.  Given the exact question [insert question], prioritize retaining all facts,
named entities, numeric values, and regulatory terms needed to answer it fully and precisely.
  1. Preserve Named Entities and Roles
       • Always retain full names, official titles, and specific roles or statuses of individuals,
         councilors, officials, organizations, and ordinance titles without abbreviation or generalization.
       • Explicitly link each named individual or entity to their specific actions, statements, feedback, or
         sentiments (e.g., ``Mark Hersman read a letter into the record'').
       • Avoid pronouns or vague references that obscure who performed an action or made a statement.
       • Preserve group membership descriptors (e.g., ``City council members,'' not just ``Council'') to
         maintain clarity and precision.
  2. Preserve Regulatory and Numeric Details Exactly
       • Retain exact regulatory terminology such as ``distance requirements,'' ``setback distances,''
         ``buffer zones,'' ``ordinance numbers,'' and ``regulatory changes'' verbatim.
       • Preserve all numeric values and spatial constraints exactly as stated (e.g., ``1,000-foot buffer,''
         ``600-foot setback'').
       • Maintain causal and definitional relationships linking policies, ordinances, and impacts verbatim or
         nearly verbatim.
  3. Prioritize Question-Relevant Content
       • Focus compression on preserving all information that directly or indirectly answers the given
         question.
       • Include all named entities, actions, regulatory terms, numeric details, and causal links relevant to
         the question.
       • If uncertain, err on the side of inclusion to avoid omitting critical facts.
  4. Compression Strategy
       • Compress aggressively by removing redundant procedural politeness, filler, and irrelevant details.
       • Shorten words, drop articles/prepositions, and use compact notation (e.g., numerals, ranges) only
         when clarity is maintained.
       • Never omit, alter, or paraphrase critical facts, semantic roles, official names, legal terms,
         procedural language, or length descriptors.
       • Avoid vague summarization, ambiguous phrasing, or partial references that reduce clarity or
         introduce confusion.
       • Do not introduce new phrases, synonyms, or inferred roles that could mislead or confuse downstream
         QnA.
  5. Preserve Speaker--Action Associations
       • Always preserve speaker identities and explicitly associate them with their utterances or actions
         related to the question.
       • Do not confuse, omit, or replace speaker names or roles, especially those linked to key points or
         next steps.
  6. Formatting and Clarity
       • Format key facts clearly and explicitly, preferably as concise, structured statements or bullet
         points.
       • Group related facts densely to maximize information per token while maintaining clarity.
       • Label speakers with full names and exact roles to aid downstream extraction.
  7. Procedural and Contextual Details
       • Preserve procedural outcome phrases verbatim (e.g., ``motion carries,'' ``motion denied''),
         including tense and modality.
       • Include relevant procedural context and concise clarifications when they support accurate
         understanding.
  8. Self-Review and Verification
       • After compression, perform a mandatory self-review to verify that all answer-relevant facts, named
         entities, official titles, procedural outcomes, dates, jurisdictions, sentiments, goals, length
         descriptors, numeric values, and causal relationships remain present, unambiguous, and verbatim or
         nearly verbatim.
       • Confirm the compressed text fully supports accurate question answering.
       • Reinsert any missing critical information before finalizing.
  9. Length Flexibility
       • Prioritize factual completeness and semantic fidelity over strict brevity.
       • It is acceptable to slightly exceed length limits to preserve critical information and question
         relevance.
 10. Examples
       • Good:  ``Vice Mayor Susan Lowenthal supported the housing initiative.''   Bad:  ``Staff supported
         the initiative.''
       • Good:  ``Distance requirement:  1,000-foot buffer from schools.''   Bad:  ``Regulations about
         distance.''
       • Good:  ``Mark Hersman read a letter into the record.''   Bad:  ``A resident read a letter.''
       • Good:  ``Councilor Kenzie Bok provided feedback on promotional practices.''   Bad:  ``Councilor Bok
         spoke.''
       • Good:  ``Ordinance 0259 establishes Chief Diversity Officer role.''   Bad:  ``Ordinance 0259.''
       • Good:  ``Motion carries.''   Bad:  ``Motion passed.''
       • Good:  ``Dist 3--8 voted yes.''   Bad:  ``All districts voted yes.''
       • Good:  ``Alicia Flores, victim of city ordinance, spoke about evictions.''   Bad:  ``Alicia Flores
         advocated against evictions.''
       • Good:  ``Jordan Winn, 2nd District resident and S.O.S. Board member, spoke about enforcement of
         Ordinance 1482.''   Bad:  ``A resident asked about next steps.''

Remember, the compressed transcript is the sole context for a QnA model to answer the question.  Omitting
or distorting key facts, named entities, or attributions will cause incorrect answers and reduce
evaluation scores.  Prioritize semantic fidelity and question relevance, even if this requires slightly
exceeding length limits.  Compression quality will be iteratively improved based on feedback; prioritize
precision, completeness, semantic fidelity, and question relevance accordingly.  Preserve all relevant
answer information verbatim or with minimal paraphrasing to ensure downstream accuracy.
```

## C  CMPRSR DETAILS

### C.1  GRPO TRAINING DETAILS

We fine-tune the compressor with Grouped Relative Preference Optimization (GRPO) on a 10k-example subset of the SFT training split. For each input, target compression rate is uniformly distributed with $r \sim \mathcal{U}(0.1, 0.7)$.

Training runs on two NVIDIA A100 GPUs: one GPU performs optimization, and the other handles rollout generation and CE computation, decoupling sampling from updates.

**Key hyperparameters.**  We train for two epochs with AdamW (8-bit) and a linear schedule with warmup. Mixed precision uses `bf16`. Per-device batch size is 8 with gradient accumulation of 16 (effective batch $\approx 128$ sequences per optimizer step on the training GPU).

| Setting | Value |
| --- | --- |
| Training examples | 10,000 (subset of SFT train) |
| Rollouts per input | $G = 4$ |
| Epochs | 2 |
| Optimizer | AdamW (8-bit) |
| Learning rate | $5 \times 10^{-6}$ |
| Weight decay | 0.01 |
| Scheduler / Warmup | Linear / 10% |
| Per-device train batch | 8 |
| Gradient accumulation | 16 |
| Precision | `bf16` |
| Max grad norm | 1.0 |

### C.2  DPO AND ORPO

We apply Direct Preference Optimization (DPO) to learn compression policies under a fixed token budget. Each training sample contains a prompt (with the conditions seen before) with the original chunk (extracted from MeetingBank) and a pair of candidate compressions: a *chosen* output produced by a stronger, teacher model (`gpt-4.1-mini`) and a rejected output produced by a weaker, student model (`gpt-4.1-nano`). To make sure that for each chunk the teacher-produced one contains higher quality compression, we calculate the **BERT-F1** score between the compressed and the original chunks and we select the triplets only if the teacher has produced higher F1-scored compressions. We evaluate DPO under two initializations (i) from the vanilla model and (ii) from the SFT-initialized model. Should we mention testing beta hyperparams?

We also evaluate Odds Ratio Preference Optimization (ORPO), a reference-free variant that simplifies preference optimization to a single-model setup while retaining the same paired (chosen/rejected) data. ORPO removes the need for an explicit reference model and instead adjusts the policy to increase the odds of the chosen response relative to the rejected one. As with DPO, we evaluate ORPO under two initializations—(i) from the base model and (ii) from the SFT-initialized model using the same setup to enable like-for-like comparisons

### C.3  TRAINING DATA GENERATION

Figure 7 shows the data distribution in the dataset generated to train the SFT backbone. The blue plot shows the original distribution which is favouring  0.3 compression ratio. To mitigate this, two techniques are applied: yellow showcases the downsampled version of the dataset, where we applied a cut at 150 samples in each bin, while the green showcases the oversampled variant.

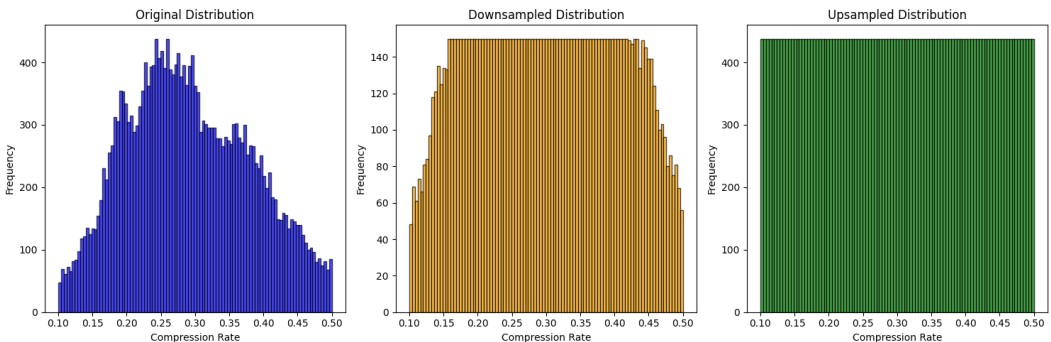

Figure 7: Data distribution

## D EXPERIMENTAL RESULTS

### D.1 QA BASED EVALUATION

We present all results from benchmarking our model on QA tasks. For this, we use GSM8K, a mathematical reasoning dataset where each instance consists of a problem statement and its corresponding ground-truth answer. For LongBench, we evaluate on three tasks: TriviaQA, NarrativeQA, and Qasper. TriviaQA is a few-shot QA task, while the other two are single-document QA tasks. For MeetingBank, we generated QA pairs from the original transcripts, which we consider as the ground truth. The results show that CMPRSR model consistently outperforms baselines, particularly under aggressive compression ratios (e.g., 0.1). On LongBench, SFT performs well at higher compression ratios, but our method remains competitive and achieves strong overall performance.

### D.2 SUMMARY BASED EVALUATION

We also evaluate our approach on summarization tasks using the **MeetingBank** summarization dataset, as well as three splits from the LongBench benchmark: *GovReport*, *QMSum*, and *Multi-News*. The primary evaluation metric we use is **BERTScore**, which allows us to quantify the similarity between the generated summaries and the reference texts. Because the score range of BERTScore is very narrow, we report all results up to four decimal places for clarity and precision, ensuring that even small differences in performance are visible and interpretable.

The CMPRSR model achieves dominant performance on MeetingBank, benefiting from its in-domain characteristics, while also generalizing well to the LongBench benchmarks. It is important to note that BERTScore is correlated with the target compression rate: as compression becomes more aggressive, BERTScore tends to decrease, since some information is inevitably lost, even if it is not critical. This explains why the SFT model performs better than CMPRSR on LongBench at target ratios of 0.3 and 0.5. In these settings, CMPRSR compresses more aggressively than required, leading to a stronger penalty and lower scores compared to SFT.

### D.3 PLOTS

### D.4 CROSS ENTROPY ANALYSIS FOR GEMMA-3

Table 6: Comparison of models across GSM8K, LongBench, and MB datasets.

| CR | Models | GSM8K | | triviaqa | | narrativeqa | | qasper | | MeetingBank QA | |
|---|---|---|---|---|---|---|---|---|---|---|---|
| | | $\Delta_{CR}$ | EM | $\Delta_{CR}$ | Accuracy | $\Delta_{CR}$ | Accuracy | $\Delta_{CR}$ | Accuracy | $\Delta_{CR}$ | Accuracy |
| 0.1 | Lingua 2 | 0.02 | 0.67 | -0.01 | 31.00 | -0.03 | 5.00 | 0.00 | 14.00 | -0.01 | 16.70 |
| | gpt-4.1-mini | 0.20 | 39.33 | 0.10 | 30.00 | 0.10 | 10.00 | 0.15 | 20.00 | 0.07 | 18.27 |
| | Qwen3-4b | 0.36 | 63.00 | 0.06 | 28.00 | 0.10 | 13.00 | 0.12 | 19.00 | 0.06 | 14.89 |
| | SFT | 0.28 | 54.33 | 0.05 | 23.23 | 0.08 | 12.00 | 0.09 | 17.00 | 0.07 | 17.96 |
| | GRPO | 0.24 | 7.33 | 0.00 | 29.90 | -0.02 | 7.00 | 0.01 | 16.00 | 0.02 | 17.26 |
| | CMPRSR | **-0.01** | **12.67** | **-0.02** | **32.00** | **-0.02** | **11.00** | **-0.01** | **16.00** | **-0.01** | 20.05 |
| 0.3 | Lingua 2 | 0.02 | 12.33 | 0.00 | 30.21 | -0.02 | 11.00 | **0.00** | **18.00** | -0.03 | 32.70 |
| | gpt-4.1-mini | 0.17 | 82.00 | 0.14 | 28.00 | 0.12 | 11.00 | 0.14 | 18.00 | 0.00 | 27.61 |
| | Qwen3-4b | 0.24 | 75.00 | -0.03 | 28.00 | **-0.01** | **14.00** | 0.01 | 18.00 | -0.07 | 18.69 |
| | SFT | 0.20 | 78.00 | 0.07 | 32.32 | 0.09 | 15.00 | 0.10 | 18.00 | 0.02 | 26.38 |
| | GRPO | 0.08 | 13.33 | 0.12 | 31.31 | -0.01 | 10.10 | 0.06 | 15.00 | -0.02 | 25.40 |
| | CMPRSR | **-0.05** | **35.33** | **-0.01** | **31.31** | -0.02 | 10.00 | 0.01 | 19.00 | **-0.01** | **33.20** |
| 0.5 | Lingua 2 | 0.00 | 49.33 | -0.01 | 21.43 | -0.03 | 12.00 | 0.00 | 18.00 | -0.03 | 41.14 |
| | gpt-4.1-mini | 0.06 | 87.00 | 0.00 | 29.29 | -0.03 | 16.00 | 0.00 | 18.00 | -0.17 | 28.36 |
| | Qwen3-4b | 0.05 | 76.67 | -0.20 | 28.28 | -0.15 | 13.27 | -0.14 | 16.00 | -0.24 | 20.22 |
| | SFT | 0.07 | 83.33 | **-0.01** | **27.00** | **-0.01** | **16.16** | **0.00** | **19.00** | -0.10 | 30.87 |
| | GRPO | -0.10 | 15.00 | 0.09 | 31.58 | -0.11 | 12.12 | 0.00 | 15.00 | -0.11 | 28.95 |
| | CMPRSR | **-0.12** | **62.33** | 0.03 | 29.59 | -0.01 | 12.00 | 0.06 | 19.00 | **-0.01** | **41.95** |

Table 7: Comparison of models across MB (Summary) and LongBench datasets.

| CR | Models | MeetingBank Summary | | GovReport | | QMSum | | MultiNews | |
|---|---|---|---|---|---|---|---|---|---|
| | | $\Delta_{CR}$ | BertScore | $\Delta_{CR}$ | Bert-F1 | $\Delta_{CR}$ | Bert-F1 | $\Delta_{CR}$ | Bert-F1 |
| 0.1 | Lingua 2 | -0.01 | 0.8641 | 0.00 | 0.8516 | -0.03 | 0.8598 | -0.01 | 0.8579 |
| | gpt-4.1-mini | 0.07 | 0.8865 | 0.13 | 0.8788 | 0.06 | 0.8643 | 0.06 | 0.8868 |
| | Qwen3-4b | 0.06 | 0.8629 | 0.11 | 0.8467 | 0.12 | 0.8625 | 0.05 | 0.8774 |
| | SFT | 0.07 | 0.8841 | 0.08 | 0.8748 | 0.04 | 0.8662 | 0.06 | 0.8853 |
| | GRPO | 0.02 | 0.8461 | 0.00 | 0.8372 | -0.03 | 0.8445 | 0.02 | 0.8389 |
| | CMPRSR | **-0.01** | **0.8773** | **-0.01** | **0.8631** | **-0.04** | **0.8641** | **-0.01** | **0.8802** |
| 0.3 | Lingua 2 | -0.03 | 0.8913 | **-0.01** | **0.8746** | -0.02 | 0.8686 | -0.03 | 0.8886 |
| | gpt-4.1-mini | 0.00 | 0.8989 | 0.11 | 0.8843 | 0.07 | 0.8707 | 0.03 | 0.8936 |
| | Qwen3-4b | -0.07 | 0.8823 | 0.00 | 0.8699 | -0.01 | 0.8618 | -0.08 | 0.8823 |
| | SFT | 0.02 | 0.8971 | 0.08 | 0.8820 | -0.01 | 0.8718 | 0.01 | 0.8913 |
| | GRPO | -0.02 | 0.8641 | 0.04 | 0.8611 | -0.08 | 0.8581 | 0.01 | 0.8616 |
| | CMPRSR | **-0.01** | **0.9009** | 0.02 | 0.8841 | **-0.09** | **0.8715** | -0.03 | **0.8949** |
| 0.5 | Lingua 2 | -0.03 | 0.9020 | -0.02 | 0.8838 | -0.01 | 0.8770 | **-0.04** | **0.9072** |
| | gpt-4.1-mini | -0.17 | 0.8975 | **-0.04** | **0.8863** | -0.06 | 0.8748 | -0.12 | 0.8960 |
| | Qwen3-4b | -0.24 | 0.8868 | -0.17 | 0.8747 | -0.16 | 0.8647 | -0.25 | 0.8849 |
| | SFT | -0.10 | 0.8965 | -0.03 | 0.8870 | -0.10 | 0.8742 | -0.10 | 0.8945 |
| | GRPO | -0.11 | 0.8704 | -0.01 | 0.8626 | -0.18 | 0.8604 | -0.08 | 0.8657 |
| | CMPRSR | **-0.01** | **0.9083** | 0.07 | 0.8908 | **-0.14** | **0.8797** | -0.05 | 0.9047 |

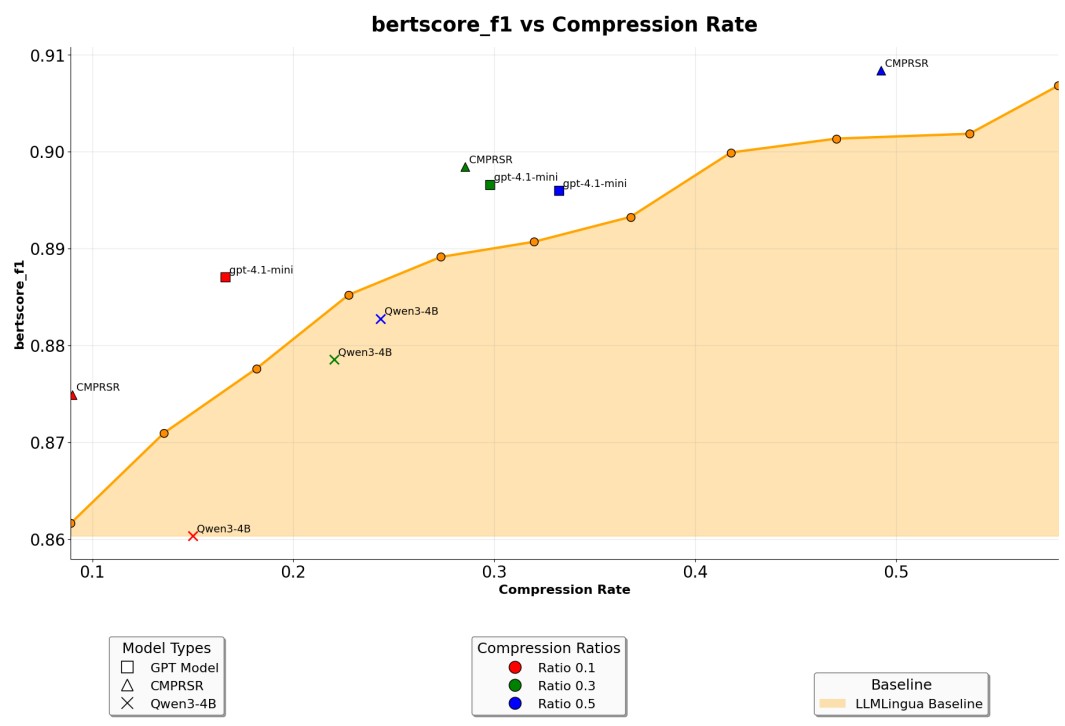

Figure 8: Comparing CMPRSR with different baselines.

Table 8: Updated Results Table

| CR | Models | Question Score | | | | | | Summary Similarity | | | |
| | | GSM8K | | LongBench | | MeetinBank | | MeetinBank | | LongBench | |
| | | $\Delta_{CR}$ | EM | $\Delta_{CR}$ | Accuracy | $\Delta_{CR}$ | Accuracy | $\Delta_{CR}$ | Bert-F1 | $\Delta_{CR}$ | Bert-F1 |
| 0.1 | Lingua 2 | 0.02 | 0.67 | -0.01 | 16.20 | -0.01 | 16.70 | -0.01 | 0.8630 | -0.01 | 0.8560 |
| | TG-gpt-4.1-mini | 0.05 | 27.33 | 0.11 | 20.82 | 0.07 | 25.05 | 0.07 | 0.8874 | 0.08 | 0.8796 |
| | TG-Qwen3-4b | 0.35 | 50.67 | 0.13 | 20.22 | 0.16 | 23.02 | 0.16 | 0.8885 | 0.15 | 0.8787 |
| | TG-SFT | 0.14 | 27.33 | 0.04 | 19.71 | 0.03 | 15.30 | 0.03 | 0.8804 | 0.03 | 0.8718 |
| 0.3 | Lingua 2 | 0.02 | 14.00 | -0.01 | 20.88 | -0.03 | 33.22 | -0.03 | 0.8897 | -0.02 | 0.8769 |
| | TG-gpt-4.1-mini | 0.16 | 66.33 | 0.18 | 20.70 | 0.07 | 36.70 | 0.07 | 0.9014 | 0.16 | 0.8886 |
| | TG-Qwen3-4b | 0.30 | 69.67 | 0.06 | 20.39 | 0.05 | 29.62 | 0.05 | 0.8978 | 0.07 | 0.8822 |
| | TG-SFT | 0.17 | 71.00 | 0.06 | 20.10 | 0.00 | 26.23 | 0.00 | 0.8962 | 0.00 | 0.8803 |
| 0.5 | Lingua 2 | 0.00 | 50.33 | -0.01 | 18.07 | -0.03 | 40.65 | -0.03 | 0.9010 | -0.02 | 0.8881 |
| | TG-gpt-4.1-mini | 0.14 | 82.67 | 0.09 | 19.67 | -0.02 | 39.90 | -0.02 | 0.9039 | 0.12 | 0.8913 |
| | TG-Qwen3-4b | 0.24 | 74.67 | -0.06 | 19.76 | -0.06 | 32.78 | -0.06 | 0.9004 | -0.04 | 0.8858 |
| | TG-SFT | 0.06 | 81.67 | 0.00 | 20.83 | -0.10 | 30.81 | -0.10 | 0.9018 | -0.09 | 0.8837 |

Table 9: Comparison of models across GSM8K, LongBench, and MB datasets.

| CR | Models | GSM8K | | LongBench | | | | | | MeetingBank | |
| | | | | triviaqa | | narrativeqa | | qasper | | QA | |
| | | $\Delta_{CR}$ | EM | $\Delta_{CR}$ | Accuracy | $\Delta_{CR}$ | Accuracy | $\Delta_{CR}$ | Accuracy | $\Delta_{CR}$ | Accuracy |
|---|---|---|---|---|---|---|---|---|---|---|---|
| 0.1 | Lingua 2 | 0.02 | 0.6667 | -0.01 | 30.61 | -0.03 | 6.00 | 0.00 | 12.00 | -0.01 | 16.70 |
| | TG-gpt-4.1-mini | 0.05 | 27.33 | 0.10 | 31.31 | 0.10 | 11.55 | 0.14 | 19.60 | 0.07 | 25.05 |
| | TG-Qwen3-4b | 0.35 | 50.67 | 0.10 | 29.66 | 0.13 | 13.00 | 0.16 | 18.00 | 0.16 | 23.02 |
| | TG-SFT | 0.14 | 27.33 | 0.03 | 30.00 | 0.04 | 14.14 | 0.06 | 15.00 | 0.03 | 15.30 |
| 0.3 | Lingua 2 | 0.02 | 14.00 | -0.00 | 32.65 | -0.02 | 12.00 | 0.00 | 18.00 | -0.03 | 33.22 |
| | TG-gpt-4.1-mini | 0.16 | 66.33 | 0.17 | 29.59 | 0.18 | 14.00 | 0.18 | 18.50 | 0.07 | 36.70 |
| | TG-Qwen3-4b | 0.30 | 69.67 | 0.05 | 30.16 | 0.06 | 13.00 | 0.08 | 18.00 | 0.05 | 29.62 |
| | TG-SFT | 0.17 | 71.00 | 0.07 | 30.30 | 0.04 | 12.00 | 0.07 | 18.00 | -0.00 | 26.23 |
| 0.5 | Lingua 2 | -0.00 | 50.33 | -0.01 | 21.21 | -0.03 | 14.00 | -0.00 | 19.00 | -0.03 | 40.65 |
| | TG-gpt-4.1-mini | 0.14 | 82.67 | 0.09 | 27.00 | 0.09 | 14.00 | 0.09 | 18.00 | -0.02 | 39.90 |
| | TG-Qwen3-4b | 0.24 | 74.67 | -0.07 | 30.16 | -0.06 | 12.63 | -0.06 | 16.50 | -0.06 | 32.78 |
| | TG-SFT | 0.06 | 81.67 | 0.01 | 30.30 | -0.01 | 14.00 | -0.01 | 18.18 | -0.10 | 30.81 |

Table 10: Comparison of models across MB (Summary) and LongBench datasets.

| CR | Models | MeetingBank | | LongBench | | | | | |
| | | Summary | | GovReport | | QMSum | | MultiNews | |
| | | $\Delta_{CR}$ | BertScore | $\Delta_{CR}$ | Bert-F1 | $\Delta_{CR}$ | Bert-F1 | $\Delta_{CR}$ | Bert-F1 |
|---|---|---|---|---|---|---|---|---|---|
| 0.1 | Lingua 2 | -0.01 | 0.86297 | -0.00 | 0.85153 | -0.03 | 0.85874 | -0.01 | 0.85776 |
| | TG-gpt-4.1-mini | 0.07 | 0.88736 | 0.11 | 0.87790 | 0.06 | 0.87025 | 0.07 | 0.89068 |
| | TG-Qwen3-4b | 0.16 | 0.88847 | 0.20 | 0.87991 | 0.13 | 0.86965 | 0.12 | 0.88641 |
| | TG-SFT | 0.03 | 0.88037 | 0.05 | 0.86962 | 0.01 | 0.86281 | 0.02 | 0.88289 |
| 0.3 | Lingua 2 | -0.03 | 0.88967 | -0.01 | 0.87356 | -0.02 | 0.86904 | -0.03 | 0.88819 |
| | TG-gpt-4.1-mini | 0.07 | 0.90138 | 0.19 | 0.88918 | 0.17 | 0.87675 | 0.12 | 0.89974 |
| | TG-Qwen3-4b | 0.05 | 0.89782 | 0.13 | 0.88325 | 0.01 | 0.86989 | 0.06 | 0.89335 |
| | TG-SFT | -0.00 | 0.89624 | 0.05 | 0.87981 | -0.06 | 0.87068 | -0.01 | 0.89031 |
| 0.5 | Lingua 2 | -0.03 | 0.90100 | -0.02 | 0.88222 | -0.01 | 0.87639 | -0.04 | 0.90581 |
| | TG-gpt-4.1-mini | -0.02 | 0.90386 | 0.17 | 0.89322 | 0.13 | 0.87860 | 0.06 | 0.90194 |
| | TG-Qwen3-4b | -0.06 | 0.90039 | 0.01 | 0.88656 | -0.08 | 0.87385 | -0.04 | 0.89690 |
| | TG-SFT | -0.10 | 0.90176 | -0.03 | 0.88434 | -0.13 | 0.87232 | -0.12 | 0.89447 |

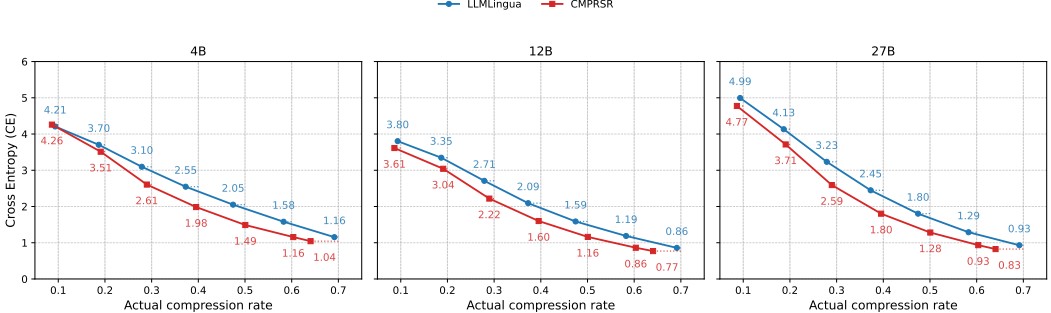

Figure 9: Cross-entropy (CE) on the summarization task for the Gemma-3 family (4B-IT, 12B-IT, 27B-IT) under two prompt-compression methods—LLMLINGUA-2 and CMPRSR—evaluated across target compression rates. Lower CE indicates better information retention. Results are averaged over 1,000 MeetingBank evaluation samples.

Table 11: N-gram overlap statistics by target compression rate.

| Target Rate | 1-gram | | 2-gram | | 3-gram | |
|---|---|---|---|---|---|---|
| | Lingua-2 | CMPRSR | Lingua-2 | CMPRSR | Lingua-2 | CMPRSR |
| 0.1 | 0.065 | 0.065 | 0.017 | 0.024 | 0.003 | 0.008 |
| 0.2 | 0.145 | 0.141 | 0.043 | 0.055 | 0.010 | 0.021 |
| 0.3 | 0.227 | 0.229 | 0.080 | 0.101 | 0.026 | 0.044 |
| 0.4 | 0.318 | 0.327 | 0.137 | 0.166 | 0.057 | 0.085 |
| 0.5 | 0.420 | 0.431 | 0.219 | 0.246 | 0.110 | 0.145 |
| 0.6 | 0.531 | 0.535 | 0.327 | 0.343 | 0.196 | 0.227 |
| 0.7 | 0.644 | 0.572 | 0.454 | 0.380 | 0.315 | 0.260 |

