# OpenReview forum: "ATAC: Abstractive Token-Level Question-Agnostic Prompt Compression"
_ICLR.cc/2026/Conference — ICLR 2026 Conference Withdrawn Submission_

### Official Review · Reviewer_vc76 · 2025-10-17

**Soundness:** 3
**Presentation:** 2
**Contribution:** 2
**Rating:** 4
**Confidence:** 4

**Summary:**

This paper presents a novel, abstractive, token-level prompt compressor built on the Qwen3-4B LLM. The goal is to reduce the cost of large, black-box LLMs by compressing inputs while strictly adhering to a target compression rate. The authors benchmarked vanilla LLMs on a new suite, CompressionBench, revealing that compression rate adherence is a major weakness for off-the-shelf models. The proposed method is developed via SFT and GRPO), and demonstrates state-of-the-art performance, particularly in terms of fidelity and stable CR control, outperforming extractive SOTA LLMLingua-2 and vanilla abstractive methods.

**Strengths:**

The paper tackles a practical problem: reducing LLM inference costs without sacrificing performance. The focus on question-agnostic compression is well-motivated—many real-world pipelines (e.g., RAG with pre-compressed documents, batch analysis of meeting transcripts) benefit from reusable compressed contexts.

The use of TextGrad to automate prompt optimization could be a strength, which eliminates the need for manual prompt engineering, allowing for more systematic and efficient refinement of the compression process.

The experiment evaluation is thorough, especially the detailed benchmarking in terms of compression rate adherence rather than only evaluating the downstream task performance.

The paper is well-written, the authors well explained complex concepts like TextGrad, GRPO reward, and compression metrics. The method is introduced step-by-step, and the results are presented with sufficient detail to support the claims made.

**Weaknesses:**

First, the claimed API cost savings are presented under the implicit assumption that compression is “free.” However, deploying a fine-tuned 4B compressor incurs nontrivial compute and latency overhead—especially compared to lightweight extractive baselines like LLMLingua-2 (0.55B). The paper briefly acknowledges this in Section 4.4 but It would be more beneficial for the authors to analyze: how many downstream LLM calls on a compressed document are needed to make the total cost or latency (compression + inference) lower than using the original prompt.

The reward design in GRPO appears somewhat ad-hoc. The authors mention abandoning a QA-based quality reward because the compressor started “outputting many possible answers” instead of compressing—but this is puzzling. Since the compression is question-agnostic, the compressor never sees the question, why would it generate many answers?

The Quality reward formula appears to be incorrect. the denominator likely lacks a negative sign, which could lead to negative or undefined values (if the denominator is zero or negative).

The CompressionBench proposed by the authors seems to be primarily a simple ensemble of benchmarks like GSM8K, MeetingBank, and LongBench, which are relatively simple for current powerful LLMs. The authors do not evaluate more challenging, recent benchmarks such as InfiniteBench, Ruler, and LongBenchv2.

**Questions:**

As mentioned in the weaknesses section, since the authors focus on question-agnostic compression, the model does not have access to the specific question. Why, then, does the compressor generate "many possible answers"? Furthermore, why would a summary-based reward help mitigate this issue? A deeper explanation would be valuable.

---

### Official Review · Reviewer_ETc9 · 2025-10-28

**Soundness:** 3
**Presentation:** 2
**Contribution:** 2
**Rating:** 2
**Confidence:** 4

**Summary:**

This paper introduces Cmprsr, a model for abstractive, question-agnostic prompt compression.

**Strengths:**

- The paper presents a benchmark evaluating 25 LLMs for prompt compression.

- The authors evaluate their method across multiple datasets (MeetingBank, LongBench, GSM8k) and two downstream tasks (summarization and QA).

**Weaknesses:**

- The paper only compares against LLMLingua-2, a prompt compression method from 2024, and lacks a comparison with newer methods from 2025 (such as CoLoR[1] and DAC[2]).

- The paper compares Cmprsr, which is based on a 4B parameter model (Qwen3-4B), against LLMLingua-2, which uses a 0.55B parameter model. While the authors acknowledge this size difference in the discussion, they still use it as a primary baseline for claiming SOTA performance.

- The proposed Cmprsr is autoregressive, which inherently incurs more latency than the extractive LLMLingua-2. The authors' main rebuttal is that compression can be pre-computed, which is not a valid assumption for many real-time use cases.

[1] Minju Seo, et al., Efficient Long Context Language Model Retrieval with Compression, ACL 2025.

[2] Yi Zhao, et al., DAC: A Dynamic Attention-aware Approach for Task-Agnostic Prompt Compression, ACL 2025.

**Questions:**

- Line 347 and Line 378 refers to Table 8, but the results being discussed are in Table 2 on page 7.

- The title introduces ATAC, but this acronym is never defined or used again in the paper.

---

### Official Review · Reviewer_NTtu · 2025-11-01

**Soundness:** 1
**Presentation:** 1
**Contribution:** 1
**Rating:** 0
**Confidence:** 4

**Summary:**

This work introduces **Cmprsr**, a *Qwen-3-4B* distillation from GPT-4.1-mini via supervise fine-tuning (SFT) and group relative policy optimization (GRPO) that best adheres to user-defined compression rates (CRs).

Also introduced is a large language model (LLM)-as-a-compressor benchmark by evaluating model compression performance in terms of deviation from prompted CR ($\Delta_{CR}$), BERT-F1-score, and question answering (QA) accuracy.

Before introducing **Cmprsr**, the authors also use TextGrad to perform system prompt optimization for GPT-4.1-mini.

To train **Cmprsr** with GRPO, a length and quality reward are used, which encourage adherance to prompted compression ratio and to information preservation respectively.

**Cmprsr** is evaluated on GSM8K, LongBench, and MeetingBank and has the best (lowest) $\Delta_{CR}$ compared to the baselines, although it is outperformed on EM (exact match?), Accuracy, and BERT-F1 in many cases.

Additional analysis is performed on the practical applicability of **Cmprsr**, as well as a comparison of the cross-entropy of prompts compressed with **Cmprsr** and LLMLingua-2.

**Strengths:**

- **Cmprsr** achieves the lowest $\Delta_{CR}$ compared to benchmarks and lower summarization cross-entropy loss compared to LLMLingua-2
- The work offers many details on hard-prompt compression

**Weaknesses:**

- This work overclaims its accomplishments: it neither "introduces a novel prompt compression paradigm" nor "presents the first comprehensive LLM-as-a-compressor benchmark"
- Uncertainty estimates are missing and the results are very similar, calling to question the validity of the conclusions
- Figure 2 is unnecessarily complicated, Figure 3 the labels of points are hard to see and an arrow goes through two of the labels
- Table 2 is misleading:
  - If $\Delta_{CR}$ in Table 2 is absolute deviation as opposed to the definition given in Section 3, then it should not be negative anywhere in Table 2
  - **Cmprsr** results are not the best in many cases, but are bolded in each column
  - EM (probably exact match) is not defined in the text
- **Cmprsr** is claimed to generate "SOTA level compressions" under $\Delta_{CR} \leq 0$ while maximizing task performance, but:
  - This objective is never mentioned earlier in the paper
  - No rigorous comparison is made besides noting that no baselines achieve negative $\Delta_{CR}$
- Table 8 is referenced in the main body of the text, but not included, while Table 2 is included, but not referenced.
- Many missing references (GRPO/DeepSeek, Mistral, Gemma, Flan, BERT, etc.)

**Questions:**

- Considering how similar many of the reported results are, what is the uncertainty in each of the measurements?
- How does the SFT + GRPO combination on other models with available weights besides Qwen3-4B?
- How were the target compression rates (0.1, 0.3, 0.5) chosen?

---

### Official Review · Reviewer_rq5h · 2025-11-01

**Soundness:** 2
**Presentation:** 3
**Contribution:** 2
**Rating:** 2
**Confidence:** 4

**Summary:**

The paper proposes ATAC, a framework for task-agnostic prompt compression that aims to reduce LLM context length while preserving downstream task performance. The approach first uses TextGrad, a natural-language “pseudo-gradient” prompt-evolution procedure, to iteratively improve a system prompt that instructs a strong teacher model (e.g., GPT-4.1-mini) to produce high-quality abstractive compressions under strict token budgets. These compressed outputs are then used to distill the compression behavior into a small student model (Qwen-4B) via supervised fine-tuning, followed by GRPO-based reinforcement learning to further enforce both information retention and exact compression-ratio adherence. Experiments on MeetingBank and LongBench show that this distilled compressor (“Cmprsr-Qwen-4B”) can outperform vanilla LLMs and LLMLingua-2 under tight token budgets, enabling cheaper and faster inference pipelines while maintaining competitive QA accuracy.

**Strengths:**

1. The paper offers a fresh perspective on context compression by explicitly formulating it as an abstractive, task-agnostic problem rather than selective extraction, and operationalizes this with a novel meta-prompt optimization framework (TextGrad) combined with policy distillation and RL fine-tuning.

2. The experimental pipeline is thoughtful and well engineered, showing a clear progression from prompt discovery to distillation and reinforcement learning, with consistent gains over existing prompt-based compressive baselines and strong evidence that the learned compressor generalizes across domains.

**Weaknesses:**

1. The empirical comparisons are narrower than expected for a paper proposing a universal compression framework. The evaluation focuses mainly on LLMLingua baseline, while omitting other compression work, such as DAC [1] and Selective-Context [2]. I think these methods target the same problem (i.e., reducing context size without sacrificing downstream utility). Including them, or at least discussing their relative positioning, would significantly clarify the scope of ATAC’s contribution and strengthen the case for generality extractive baselines.

2. While the paper emphasizes question-agnostic abstraction, the current evaluation protocol may not fully stress-test this claim. The QA supervision and the selected benchmarks primarily test single-hop factual recall and short-range contextual grounding. To convincingly demonstrate that ATAC preserves information for arbitrary downstream tasks rather than simply optimizing for the QA signals it is trained with, richer and more diverse stress scenarios I think would be informative. e.g., multi-hop reasoning task or multi-hop reasoning task. Broadening the evaluation in this direction would help ensure the method truly captures domain-general compressive principles.

---

[1] DAC: A Dynamic Attention-aware Approach for Task-Agnostic Prompt Compression, ACL 2025

[2] Compressing Context to Enhance Inference Efficiency of Large Language Models, EMNLP 2023

**Questions:**

1. A first question I had concerns the supervision strategy for training the student compressor. Since the pipeline ultimately distills into Qwen, why not generate the strongest possible compressed output for each input instance (e.g., searching for a per-instance optimal compression) and use those to train the student model? I understand the motivation to learn a generalizable compression policy rather than memorize local optima, but if the goal is to produce the strongest deployable compressor, it would be helpful to further clarify why learning a single, global optimized prompt is preferable to distilling from instance-level best compressions.

2. A second question relates to the “question-agnostic” framing. The current evaluation primarily focuses on single-turn tasks. Would the method behave differently in multi-hop or conversational settings, where relevant information may span long-range dependencies or evolve as the dialog progresses? Additionally, would the optimal reward design need to change under such scenarios, and did the authors consider alternative reward formulations for tasks beyond single-turn QA?

---

### Note · Authors · 2026-01-06

**Comment:**

Thank you to all the reviewers for their feedback. We've decided to withdraw the paper.

**Withdrawal Confirmation:**

I have read and agree with the venue's withdrawal policy on behalf of myself and my co-authors.